# Selective Aggregation for Low-Rank Adaptation in Federated Learning

**Pengxin Guo**[1] **Shuang Zeng**[1] **Yanran Wang**[2] **Huijie Fan**[3] **Feifei Wang**[1,4] **Liangqiong Qu**[1]*
[1] The University of Hong Kong   [2] Stanford University
[3] Shenyang Institute of Automation, Chinese Academy of Sciences
[4] Materials Innovation Institute for Life Sciences and Energy (MILES), HKU-SIRI
{guopx,zengsh9}@connect.hku.hk, joycewyr@stanford.edu,
fanhuijie@sia.cn, ffwang@eee.hku.hk, liangqqu@hku.hk

## Abstract

We investigate LoRA in federated learning through the lens of the asymmetry analysis of the learned $A$ and $B$ matrices. In doing so, we uncover that $A$ matrices are responsible for learning general knowledge, while $B$ matrices focus on capturing client-specific knowledge. Based on this finding, we introduce Federated Share-A Low-Rank Adaptation (FedSA-LoRA), which employs two low-rank trainable matrices $A$ and $B$ to model the weight update, but only $A$ matrices are shared with the server for aggregation. Moreover, we delve into the relationship between the learned $A$ and $B$ matrices in other LoRA variants, such as rsLoRA and VeRA, revealing a consistent pattern. Consequently, we extend our FedSA-LoRA method to these LoRA variants, resulting in FedSA-rsLoRA and FedSA-VeRA. In this way, we establish a general paradigm for integrating LoRA with FL, offering guidance for future work on subsequent LoRA variants combined with FL. Extensive experimental results on natural language understanding and generation tasks demonstrate the effectiveness of the proposed method. Our code is available at https://github.com/Pengxin-Guo/FedSA-LoRA.

## 1 Introduction

Large Language Models (LLMs) trained on large amounts of text, referred to as Pre-trained Language Models (PLMs), have become a cornerstone of Natural Language Processing (NLP) (Brown, 2020; Touvron et al., 2023; Achiam et al., 2023; Chowdhery et al., 2023). Typically, to adapt PLMs for specific tasks or enhance accuracy in real-world scenarios, fine-tuning PLMs on task-specific data is often needed. However, in many real-world applications, data is distributed across different institutions, and data sharing between these entities is often restricted due to privacy and regulatory concerns. Federated Learning (FL) (McMahan et al., 2017; Li et al., 2020a; Zhang et al., 2021; Kairouz et al., 2021), which utilizes collaborative and decentralized training of models across multiple institutions without sharing personal data externally, offers a promising solution to this challenge.

Despite its promise, fine-tuning PLMs in an FL system is challenging due to the high computational and storage demands on local clients and the communication overhead involved. To enable fine-tuning of PLMs in an FL system with limited resources, various Parameter-Efficient Fine-Tuning (PEFT) techniques have been explored. These include adapter-tuning-based methods (Houlsby et al., 2019; Zhang et al., 2024b), prompt-tuning-based methods (Li & Liang, 2021; Guo et al., 2023b; Che et al., 2023; Guo et al., 2023a; Qiu et al., 2024; Li et al., 2024; Deng et al., 2024; Sun et al., 2024a; Cui et al., 2024; Cao et al., 2024), and LoRA-based methods (Hu et al., 2022; Yi et al., 2023; Liu et al., 2023; Yang et al., 2024; Qi et al., 2024; Cho et al., 2023; Byun & Lee, 2024; Chen et al., 2024; Sun et al., 2024b; Lin et al., 2024; Wu et al., 2024; Wang et al., 2024b). Among these, LoRA-based methods have become increasingly popular, leveraging the assumption that over-parameterized models have a low intrinsic dimension (Li et al., 2018; Aghajanyan et al., 2020). A pre-trained model can be shared and utilized to create multiple small LoRA modules tailored for

---

*Corresponding author.

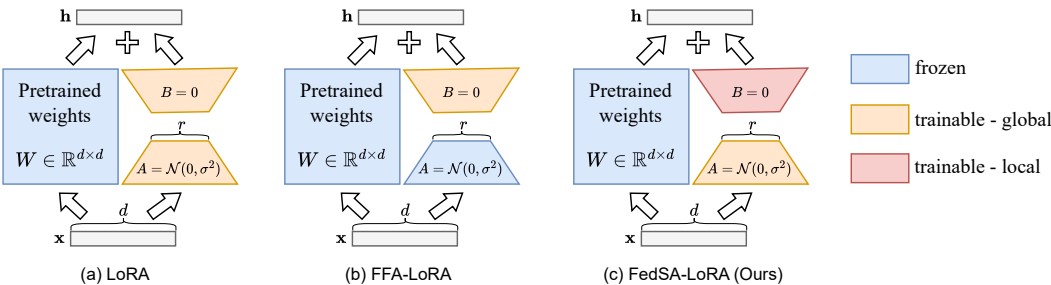

Figure 1: The illustration of (a) LoRA, (b) FFA-LoRA, and (c) FedSA-LoRA. In LoRA, both $A$ and $B$ matrices are trainable and shared with the server for aggregation. In FFA-LoRA, only $B$ matrices are trainable and shared with the server for aggregation, while $A$ matrices are fixed after initialization. In FedSA-LoRA, both $A$ and $B$ matrices are trainable, but only $A$ matrices are shared with the server for aggregation while $B$ matrices are kept locally.

different tasks, making them more effective and flexible. Moreover, this simple design allows us to merge the trainable matrices with the frozen weights during deployment, introducing no inference latency. Given these advantages, we focus on LoRA-based methods in this work.

However, aggregating LoRA matrices $A$ and $B$ in FL setting poses a key problem. Directly aggregating the $A$ and $B$ matrices on the server and then broadcasting them to each client may introduce aggregation errors. Specifically, in an FL task with $m$ clients, each client's model update is represented by two low-rank matrices $A_i$ and $B_i$ introduced by LoRA. After server aggregation and broadcast, the model update of each client is:

$$\frac{1}{m}(B_1 + B_2 + \cdots + B_m)\frac{1}{m}(A_1 + A_2 + \cdots + A_m), \tag{1}$$

which is different from the "ideal" model update, i.e., $\frac{1}{m}(B_1 A_1 + B_2 A_2 + \cdots + B_m A_m)$. [1]

To solve this problem, some methods have been explored (Sun et al., 2024b; Wang et al., 2024b). For example, Sun et al. (2024b) propose Federated Freeze-A LoRA (FFA-LoRA), which freezes the $A$ matrices and only updates and aggregates the $B$ matrices, as illustrated in Figure 1(b). Thus, the local update of each client under FFA-LoRA is $\frac{1}{m}(B_1 + B_2 + \cdots + B_m)A_0$, where $A_0$ denotes the initialized and fixed weights. They point out that this term is equal to the "ideal" model update introduced by FFA-LoRA, i.e., $\frac{1}{m}(B_1 A_0 + B_2 A_0 + \cdots + B_m A_0)$. However, fixing $A$ matrices can impair the learning ability of LoRA and result in suboptimal performance (Zhang et al., 2023). Meanwhile, many works have demonstrated that a uniform model update for all clients is not optimal, especially under the non-IID scenario (Zhao et al., 2018; Zhu et al., 2021; Li et al., 2022), which motivates the development of personalized FL (T Dinh et al., 2020; Collins et al., 2021; Xu et al., 2023). To this end, we aim to explore a better way to combine LoRA and FL in this work and move beyond the constraint that the model update of each client should be the same.

To achieve this, we start by analyzing the distinct roles of the learned $A$ and $B$ matrices when combining LoRA with FL, resulting in Lemma 1. This lemma suggests that when combing LoRA with FL, $A$ matrices are responsible for learning general knowledge while $B$ matrices focus on capturing client-specific knowledge. To verify this empirically, we locally fine-tuned a RoBERTa-large model (Liu et al., 2019) with LoRA (Hu et al., 2022) on the RTE task from the GLUE benchmark (Wang et al., 2018) with three clients under different levels of data heterogeneity. The results, illustrated in Figure 2, show that the learned $A$ matrices are more similar across clients than the $B$ matrices, and with increased data heterogeneity, the similarity of $B$ matrices between different clients decreases. These results demonstrate our argument that $A$ matrices are used to learn general knowledge while $B$ matrices focus on modeling client-specific knowledge.

Based on our findings, we introduce the Federated Share-A Low-Rank Adaptation (FedSA-LoRA) method in this work. Similar to LoRA (Hu et al., 2022), we utilize two trainable low-rank matrices, denoted as $A$ and $B$, to model the weight updates during local training. However, only the $A$ matrices are shared with the server for aggregation, as illustrated in Figure 1(c). Then, the model

---

[1]Refer to Section A.3 in Appendix for more explanations about the derived aggregation errors.

update of client $i$ after server aggregation and broadcast is:

$$B_i \frac{1}{m}(A_1 + A_2 + \cdots + A_m), \tag{2}$$

where the first part $B_i$ is responsible for capturing client-specific knowledge while the second part is used to model general knowledge. By sharing the $A$ matrices that learn general knowledge with the server for aggregation, while keeping the $B$ matrices that capture client-specific knowledge locally during training, the learning abilities of LoRA combined with FL can be enhanced. Note that this method differs from previous works (Sun et al., 2024b; Wang et al., 2024b) that require each client to share a uniform model update. Instead, it allows for different model updates, placing it in the realm of personalized FL (T Dinh et al., 2020; Collins et al., 2021; Xu et al., 2023), which is more efficient under the non-IID scenario (Zhao et al., 2018; Zhu et al., 2021; Li et al., 2022).

Moreover, we delve into the relationship between the learned $A$ and $B$ matrices in other LoRA variants, such as rsLoRA (Kalajdzievski, 2023) and VeRA (Kopiczko et al., 2024). The observations, illustrated in Figures 5 and 6 in Appendix, demonstrate a similar phenomenon to LoRA. Building upon these insights, we extend our FedSA-LoRA method to these LoRA variants, resulting in FedSA-rsLoRA and FedSA-VeRA. By extending the proposed method to other LoRA variants, we establish a general paradigm for integrating LoRA with FL, offering guidance for future work on subsequent LoRA variants combined with FL.

We summarize our contributions as follows:

- We investigate the relationship between learned $A$ and $B$ matrices in LoRA and other LoRA variants (e.g., rsLoRA and VeRA) across different clients, delineating their distinct roles. Specifically, $A$ matrices are responsible for learning general knowledge, while $B$ matrices focus on capturing client-specific knowledge.

- Building upon our findings, we establish a general paradigm for integrating LoRA with FL. Specifically, we introduce Federated Share-A LoRA (FedSA-LoRA), where both $A$ and $B$ matrices are trainable, but only the $A$ matrices are shared with the server for aggregation. We then generalize the FedSA-LoRA framework to other LoRA variants, resulting in FedSA-rsLoRA and FedSA-VeRA.

- Extensive experimental results demonstrate the superiority of the proposed FedSA-LoRA, FedSA-rsLoRA, and FedSA-VeRA compared to other methods.

## 2 RELATED WORK

### 2.1 FEDERATED LEARNING

Federated Learning (FL) (McMahan et al., 2017; Li et al., 2020a; Zhang et al., 2021; Kairouz et al., 2021), a commonly used distributed learning method for tasks requiring privacy, has gained significant attention in recent years. However, its application faces challenges due to the non-IID nature of distributed datasets, resulting in accuracy discrepancies compared to centralized training. Numerous works (Li et al., 2020b; Xu et al., 2023; Yan et al., 2023; Chan et al., 2024; Xu et al., 2024; Zeng et al., 2024) have been proposed to mitigate this performance degradation, including optimizing local learning (Li et al., 2020b), optimizing server aggregation (Zeng et al., 2024), and personalized FL (Xu et al., 2023). Recently, some studies demonstrate that fine-tuning pre-trained models, especially Pre-trained Language Models (PLMs), through FL suffers less from the non-IID issue (Qu et al., 2022; Chen et al., 2023; Nguyen et al., 2023; Weller et al., 2022). The experimental results in (Weller et al., 2022) show that when applying PLMs, even the vanilla FedAvg can achieve performance comparable to centralized training. However, these large-scale PLMs usually introduce significant communication overheads in FL scenarios, leading to slow and impractical federated training in real-world applications. Additionally, local clients are often constrained by limited computational capacity and memory, making the local fine-tuning of PLMs challenging. To enable fine-tuning of PLMs in an FL system with limited resources, various Parameter-Efficient Fine-Tuning (PEFT) techniques have been explored, such as adapter-tuning-based methods (Houlsby et al., 2019; Zhang et al., 2024b), prompt-tuning-based methods (Li & Liang, 2021; Guo et al., 2023b; Che et al., 2023; Guo et al., 2023a; Qiu et al., 2024; Li et al., 2024; Deng et al., 2024; Sun et al., 2024a; Cui et al., 2024; Cao et al., 2024), and LoRA-based methods (Hu et al., 2022; Yi et al., 2023; Liu et al., 2023;

Yang et al., 2024; Qi et al., 2024; Cho et al., 2023; Byun & Lee, 2024; Chen et al., 2024; Sun et al., 2024b; Lin et al., 2024; Wu et al., 2024; Wang et al., 2024b). Among these, LoRA-based methods have become increasingly popular, leveraging the assumption that over-parameterized models have a low intrinsic dimension (Li et al., 2018; Aghajanyan et al., 2020). A pre-trained model can be shared and utilized to create multiple small LoRA modules tailored for different tasks, making them more effective and flexible. Moreover, this simple design allows us to merge the trainable matrices with the frozen weights during deployment, introducing no inference latency. Given these advantages, we focus on LoRA-based methods in this work.

## 2.2 LoRA in Federated Learning

Low-Rank Adaptation (LoRA) (Hu et al., 2022), which introduces low-rank adaptation matrices to simulate gradient updates while keeping the pre-trained model weights frozen, has recently gained significant attention due to its efficiency, effectiveness, and flexibility (Hayou et al., 2024; Liu et al., 2024; Kopiczko et al., 2024; Wang et al., 2024a). With this trait, LoRA can be utilized to mitigate the communication overhead in FL, which primarily relies on the size of model update parameters. Yi et al. (2023) propose FedLoRA, incorporating LoRA in FL to increase model fine-tuning efficiency. Liu et al. (2023) introduce DP-LoRA, ensuring differential privacy in FL for LLMs with minimal communication overhead. Yang et al. (2024) propose a dual-personalizing adapter (FedDPA), and Qi et al. (2024) introduce FDLoRA. Both adopt the similar idea where each client contains a personalized LoRA module and a global LoRA module to capture personalized and global knowledge, respectively.

Another line of such work is heterogeneous LoRA. For example, Cho et al. (2023) introduce heterogeneous LoRA, where they deploy heterogeneous ranks across clients, aggregate the heterogeneous LoRA modules through zero-padding, and redistribute the LoRA modules heterogeneously through truncation. However, this simple zero-padding strategy can make the training process unstable (Byun & Lee, 2024). To solve this issue, Byun & Lee (2024) propose a replication-based strategy for aggregating rank-heterogeneous LoRA. Chen et al. (2024) propose Rank-Based LoRA Aggregation (RBLA) that performs a weighted aggregation for heterogeneous LoRA structures. Wang et al. (2024b) introduce a stacking-based aggregation method for heterogeneous LoRA.

The most related work to ours is Federated Freeze-A LoRA (FFA-LoRA) (Sun et al., 2024b), which fixes the randomly initialized non-zero $A$ matrices and only fine-tunes the zero-initialized $B$ matrices to further halve the communication cost. However, since some matrices are fixed, the learning ability of LoRA is impaired, resulting in suboptimal performance (Zhang et al., 2023). In contrast, we propose Federated Share-A LoRA (FedSA-LoRA), where both $A$ and $B$ matrices are trainable and only the $A$ matrices are shared with the server for aggregation.

## 3 Motivating Example

**Preliminary**   Building upon the hypothesis that updates to the weights during the fine-tuning exhibit a low "intrinsic rank", LoRA (Hu et al., 2022) proposes using the product of two low-rank matrices to update the pre-trained weights incrementally. For a pre-trained weight matrix $W_0 \in \mathbb{R}^{k \times d}$, LoRA models the weight update $\Delta_W \in \mathbb{R}^{k \times d}$ utilizing a low-rank decomposition, expressed as $BA$, where $B \in \mathbb{R}^{k \times r}$ and $A \in \mathbb{R}^{r \times d}$ represent two low-rank matrices, with $r \ll min(k, d)$. During training, $W_0$ is frozen and does not receive gradient updates, while $A$ and $B$ contain trainable parameters. Consequently, the fine-tuned weight $W'$ can be represented as: $W' = W_0 + BA$. The matrix $A$ is initialized with random Gaussian distribution, while $B$ is initially set to zero, resulting in $\Delta_W = BA$ being zero at the start of training.

To analyze the role of learned $A$ and $B$ matrices, let's consider a simple example analogous to a single network layer with least-squares linear regression task. Specifically, suppose there is a pre-trained linear model weight $W_0 \in \mathbb{R}^{k \times d}$. With this model held constant, our goal is regressing $(x_t, y_t)$ pairs where $y_t$ is given by:

$$y_t = W_t x_t,$$

with $W_t = W_0 + \Delta_W$. In LoRA, the target $\Delta_W$ is modeled by a low rank update to the pre-trained $W_0$, i.e., $W' = W_0 + BA$:

$$\hat{y} = (W_0 + BA)x_t,$$

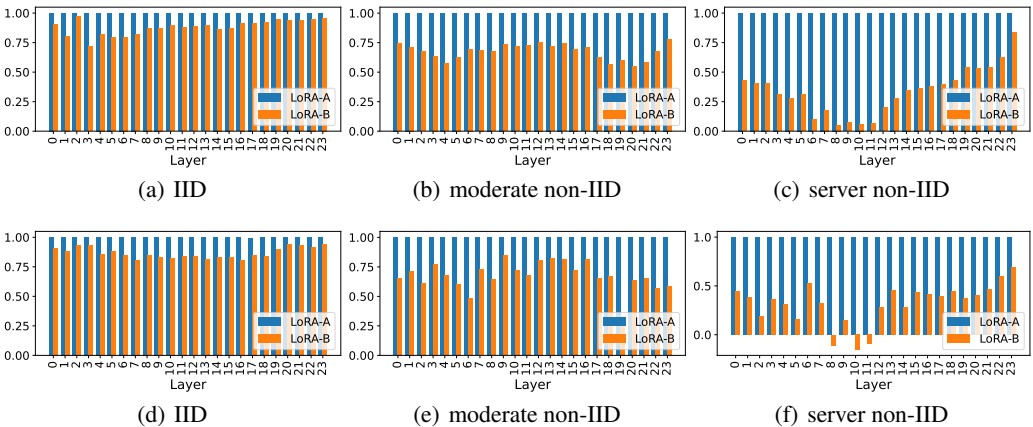

Figure 2: Mean of pairwise cosine similarity of the learned $A$ and $B$ matrices across layers of a RoBERTa model locally fine-tuned with LoRA on the RTE task, with different levels of data heterogeneity. (a)-(c): value matrices; (d)-(f): query matrices. The learned $A$ matrices are more similar across clients than the $B$ matrices, and with increased data heterogeneity, the similarity of $B$ matrices between different clients decreases.

where $B \in \mathbb{R}^{k \times r}$ and $A \in \mathbb{R}^{r \times d}$, with $r \ll min(k, d)$. Then, the least squares loss is defined on the difference between $\hat{y}$ and $y_t$:

$$\mathcal{L} = \mathbb{E}_{(x_t, y_t)}[\|y_t - (W_0 + BA)x_t\|_2^2]. \tag{3}$$

Below, we present the lemma on minimizing this loss while freezing either $A$ or $B$. The proof is provided in Section A.1 in Appendix.

**Lemma 1.** *Fine-tuning $B$ while fixing $A = Q$, with the goal of optimizing Eq. (3), yields:*

$$B^* = \Delta_W \mathbb{E}[x_t x_t^T] Q^T (Q \mathbb{E}[x_t x_t^T] Q^T)^{-1}. \tag{4}$$

*Fine-tuning $A$ while fixing $B = U$ and assuming $U^{-1}$ exists, with the goal of optimizing Eq. (3), yields:*

$$A^* = U^{-1} \Delta_W. \tag{5}$$

**Remark 1.** *From this lemma, we can conclude that the optimal solution of $A^*$ is independent of the input data distribution, while $B^*$ is related to the input data distribution captured by $\mathbb{E}[x_t x_t^T]$. This indicates that $A$ is responsible for learning general knowledge, while $B$ focuses on modeling client-specific knowledge.*

To verify this empirically, we locally fine-tune a RoBERTa-large model (Liu et al., 2019) with LoRA (Hu et al., 2022) on the RTE task from the GLUE benchmark (Wang et al., 2018) using three clients. We model an IID data distribution and two non-IID data distributions. The two non-IID distributions are modeled by a Dirichlet distribution with $\alpha = 1$ and $\alpha = 0.5$, referred to as moderate non-IID and severe non-IID. Figure 2 shows the mean pairwise cosine similarity of the learned $A$ and $B$ matrices across clients. These results indicate that the learned $A$ matrices are more similar across clients than the $B$ matrices, and with increased data heterogeneity, the similarity of $B$ matrices between different clients decreases. To demonstrate that the $A$ matrices are indeed updated, as they are similar across different clients, we further illustrate the difference between the learned and initialized $A$ matrices for each client in Figure 4 in Appendix. These results confirm that the $A$ matrices are updated. This phenomenon is consistent with previous study about the asymmetry analysis in LoRA (Zhu et al., 2024). Based on these results, we argue that $A$ matrices are responsible for learning general knowledge while $B$ matrices focus on capturing client-specific knowledge.

To demonstrate the generalizability of our findings, we further explore the relationship between the learned $A$ and $B$ matrices in other LoRA variants, such as rsLoRA (Kalajdzievski, 2023) and VeRA (Kopiczko et al., 2024). The observations, illustrated in Figures 5 and 6 in Appendix, show a similar phenomenon to LoRA. In this way, we uncover a general phenomenon when combining LoRA with FL, which serves as the foundation for our proposed method.

There are some works analyzing the asymmetry of $A$ and $B$ within LoRA in other areas (Zhu et al., 2024; Tian et al., 2024). However, our work distinctly considers the asymmetry analyses of $A$ and $B$ within LoRA in the context of federated learning, while these related papers (Zhu et al., 2024; Tian et al., 2024) can further serve as verification of the effectiveness of our method. In particular, we further analyzed different non-IID scenarios, as data heterogeneity is a significant issue in FL. Through our analysis, we not only found that the learned $A$ matrices are more similar across clients than the $B$ matrices (consistent with previous findings), but more importantly, we discovered that with increased data heterogeneity, the similarity of $B$ matrices between different clients decreases. Moreover, we further extended this asymmetry analysis to other LoRA variants, such as rsLoRA (Kalajdzievski, 2023) and VeRA (Kopiczko et al., 2024), and found similar phenomena. This generalization was previously lacking in the literature (Zhu et al., 2024; Tian et al., 2024).

## 4 OUR METHOD

### 4.1 FEDERATED SHARE-A LOW RANK ADAPTATION

Drawing from the insights of our findings, we introduce Federated Share-A Low-Rank Adaptation (FedSA-LoRA), illustrated in Figure 1(c), which utilizes two low-rank trainable matrices $A$ and $B$ to model the weight update, but only $A$ matrices are shared with the server for aggregation. Specifically, similar to LoRA (Hu et al., 2022), we employ two low-rank matrices, namely $B \in \mathbb{R}^{k \times r}$ and $A \in \mathbb{R}^{r \times d}$ with $r \ll \min(k, d)$, to model the weight update $\Delta_W \in \mathbb{R}^{k \times d}$ for a pre-trained weight matrix $W_0 \in \mathbb{R}^{k \times d}$. This approach allows us to represent the fine-tuned weight as $W_0 + BA$. During the local training process, $W_0$ is frozen and does not receive gradient updates, while $A$ and $B$ contain trainable parameters. Following LoRA (Hu et al., 2022), the matrix $A$ is initialized with a random Gaussian distribution, whereas $B$ is initially set to zero, ensuring that $\Delta_W = BA$ is zero at the start of training. Then, for global aggregation, only the $A$ matrices are shared with the server for aggregation. Once the server averages these $A$ matrices, they are broadcast to each client for the subsequent training round. By sharing the $A$ matrices that learn general knowledge with the server for aggregation, while keeping the $B$ matrices that model client-specific knowledge locally, the learning abilities of LoRA combined with FL can be enhanced.

Moreover, based on the similar phenomena observed in other LoRA variants (i.e., rsLoRA (Kalajdzievski, 2023) and VeRA (Kopiczko et al., 2024)), we extend the FedSA-LoRA method to these variants, resulting in FedSA-rsLoRA and FedSA-VeRA. Specifically, rsLoRA (Kalajdzievski, 2023) is similar to LoRA, differing only in the scaling factor. Thus, the difference between FedSA-rsLoRA and FedSA-LoRA also lies in the scaling factor. In VeRA (Kopiczko et al., 2024), the low-rank matrices $A$ and $B$ are initialized using the uniform version of Kaiming initialization, fixed, shared across all layers, and adapted with trainable scaling vectors $d$ and $b$. The $b$ vectors are initialized to zero, and the $d$ vectors are initialized with a value of 0.1. To make the notation consistent with our work, we rewrite the scaling vectors $d$ and $b$ as $A_d$ and $B_b$ to reflect the position of each scaling vector. Thus, in FedSA-VeRA, only the scaling vector $A_d$ is shared with the server for aggregation, while $B_b$ is trained locally. By extending the proposed method to other LoRA variants, we establish a general paradigm for integrating LoRA with FL, offering guidance for future work on subsequent LoRA variants combined with FL.

### 4.2 CONVERGENCE ANALYSIS

To facilitate the convergence analysis of the proposed method, we make assumptions commonly encountered in the literature (Li et al., 2020c) to characterize the smooth and non-convex optimization landscape.

**Assumption 1.** $\mathcal{L}_1, \cdots, \mathcal{L}_m$ are all L-smooth. For all $W_{i,j}$ and $W_{i,k}$:

$$\mathcal{L}_i(W_{i,k}) \leq \mathcal{L}_i(W_{i,j}) + \langle W_{i,k} - W_{i,j}, \nabla \mathcal{L}_i(W_{i,j}) \rangle_F + \frac{L}{2} \|W_{i,k} - W_{i,j}\|_F^2.$$

*Here, $m$ is the number of clients, $W_{i,j}$ and $W_{i,k}$ represent any different model weights of client $i$, $\mathcal{L}_i$ is the empirical loss on client $i$, $\nabla \mathcal{L}_i(W_{i,j})$ represents the gradient of $\mathcal{L}_i$ with respect to $W_{i,j}$, $\langle \cdot, \cdot \rangle_F$ is the Frobenius inner product, and $\| \cdot \|_F$ denotes the Frobenius norm.*

**Assumption 2.** *Let $\xi_{i,t}$ be sampled from the $i$-th client's local data uniformly at random at $t$-th training step. The expected squared norm of stochastic gradients is uniformly bounded, i.e.,*

$\mathbb{E}\|\nabla \mathcal{L}_i(W_i^{(t)}; \xi_{i,t})\|^2 \leq G^2$, *for all $i = 1, \cdots, m$ and $t = 0, \cdots, T - 1$. Here $T$ denotes the total number of every client's training steps.*

**Assumption 3.** *Let $W_i^{(t)} = W_0 + B_i^{(t)} A_i^{(t)}$ represent the model parameters for the $i$-th client at the $t$-th step. There exist constants $C_B > 0$, $C_A > 0$, $c_B > 0$, and $c_A > 0$ such that:*

$$\|B_i^{(t)}\|_F \leq C_B,$$

$$\|A_i^{(t)}\|_F \leq C_A,$$

$$\langle A_i^{(t)\top} A_i^{(t)}, \nabla \mathcal{L}_i(W_i^{(t)})^\top \nabla \mathcal{L}_i(W_i^{(t)}) \rangle_F \geq c_A \|\nabla \mathcal{L}_i(W_i^{(t)})\|_F^2,$$

$$\langle B_i^{(t)\top} B_i^{(t)}, \nabla \mathcal{L}_i(W_i^{(t)})^\top \nabla \mathcal{L}_i(W_i^{(t)}) \rangle_F \geq c_B \|\nabla \mathcal{L}_i(W_i^{(t)})\|_F^2,$$

*for all $i = 1, \cdots, m$ and $t = 0, \cdots, T - 1$.*

Assumptions 1 and 2 are widely used $L$-smoothness and bounded second moment assumption in FL papers (Li et al., 2020c; Yu et al., 2019; Basu et al., 2019). For Assumption 3, according to the definition of the Frobenius norm $\|A\|_F = \sqrt{\sum_i^m \sum_j^n |a_{ij}|^2}$, we know that the first two inequalities in Assumption 3 hold when all parameter values in $A_i^{(t)}$ and $B_i^{(t)}$ are finite. If we denote the eigenvalues of $A_i^{(t)}$ by $\{\lambda_1, \ldots, \lambda_K\}$, then the eigenvalue of $(A_i^{(t)})^\top A_i^{(t)}$ are $\{\lambda_1^2, \ldots, \lambda_K^2\}$. Similarly, if the eigenvalues of $\nabla \mathcal{L}_i(W_i^{(t)})$ are $\{\mu_1, \ldots, \mu_K\}$, then the eigenvalues of $(\nabla \mathcal{L}_i(W_i^{(t)}))^\top \nabla \mathcal{L}_i(W_i^{(t)})$ are $\{\mu_1^2, \ldots, \mu_K^2\}$. Noting that $\langle A_i^{(t)\top} A_i^{(t)}, \nabla \mathcal{L}_i(W_i^{(t)})^\top \nabla \mathcal{L}_i(W_i^{(t)}) \rangle_F = \sum_k \lambda_k^2 \mu_k^2$, and $\|\nabla \mathcal{L}_i(W_i^{(t)})\|_F^2 = \sum_k \mu_k^2$, the third inequality of Assumption 3 holds when there exist a constant $c_A$ such that $\lambda_k^2 \geq c_A, \forall k$. In other words, the third inequality of Assumption 3 holds when all the eigenvalues of $A_i^{(t)}$ are non-zero (choose $c_A = \min\{|\lambda_1|, \ldots, |\lambda_K|\}$). Similarly, the last inequality of Assumption 3 holds when all the eigenvalues of $B_i^{(t)}$ are non-zero. Then we present the convergence rate for our method, with the proof provided in Section A.2 in Appendix.

**Theorem 1.** *Let Assumptions 1, 2, and 3 hold and $L$, $G$, $C_A$, $C_B$, $c_A$, $c_B$ be defined therein. Denote $E$ as the number of local training iterations between two communication rounds. Then, for a learning rate $\eta$, we have:*

$$\frac{1}{mT} \sum_{i=1}^{m} \sum_{t=1}^{T} \mathbb{E}\left[\left\|\nabla \mathcal{L}_i(W_i^{(t)})\right\|_F^2\right] \leq \frac{2}{c_A + c_B}\sqrt{\frac{DM}{T}}, \tag{6}$$

*where $\mathcal{L}_i(W^{(0)}) - \mathcal{L}_i(W^*) \leq D, \forall i$, and $(2C_B^2 E^2 G^2 + \frac{1}{2}G^2)\eta + \frac{3}{2}\eta^4 C_A^2 C_B^2 G^4 L + (C_A C_B G^2 + \frac{3}{2}C_A^2 L G^2 + \frac{3}{2}C_B^2 L G^2 + 2C_B^4 E^2 L G^2)\eta^2 \leq M\eta^2$.*

According to Theorem 1, we can obtain an $O(\frac{1}{\sqrt{T}})$ convergence rate towards the stationary solution under smooth and non-convex conditions. This convergence rate is comparable to that of traditional FedAvg in the non-convex scenario (Yu et al., 2019)

## 5 EXPERIMENTS

In this section, we evaluate and compare the performance of the proposed method with other methods on two types of tasks: natural language understanding and natural language generation. For the natural language understanding tasks, we use the RoBERTa model (Liu et al., 2019) evaluated on the GLUE benchmark (Wang et al., 2018), including MNLI, SST2, QNLI, QQP, and RTE. For the natural language generation tasks, we employ the LLaMA model (Touvron et al., 2023) evaluated on the GSM8K dataset (Cobbe et al., 2021). Our implementation is based on the FederatedScope-LLM library (Kuang et al., 2023). The experiments for LoRA-based methods are conducted on NVIDIA GeForce RTX 4090 and 3090 GPUs, while the rsLoRA-based and VeRA-based methods are carried out on NVIDIA L40S GPUs. All experiments are performed with half-precision enabled for efficiency. For the main results in Table 1, they are based on multiple runs to report the mean and standard deviation, while other results in our paper are based on a single run.

## 5.1 NATURAL LANGUAGE UNDERSTANDING

**Model Performance.** For the natural language understanding tasks, similar to FFA-LoRA (Sun et al., 2024a), we randomly split the data across three clients for federated learning. We model a non-IID data distribution using a Dirichlet distribution with $\alpha = 0.5$, i.e., Dir (0.5). We use the pre-trained RoBERTa-large (355M) (Liu et al., 2019) from the HuggingFace Transformers library (Wolf et al., 2020) as the base model. For LoRA-based methods optimization, we adopt the SGD optimizer (Ruder, 2016) for all approaches. We set the batch size to 128, local update steps to 10, and total communication rounds to 1000, consistent across all experiments. Similar to Hu et al. (2022), we only apply LoRA to $W_q$ and $W_v$ in the attention layers in our experiments. The rank $r = 8$ and scaling factor $\alpha = 16$ are fixed for all algorithms. We report the best result from experiments run with learning rates $\eta \in \{5\text{E-}3, 1\text{E-}2, 2\text{E-}2, 5\text{E-}2, 1\text{E-}1\}$. The optimization of rsLoRA-based methods is similar to LoRA-based methods, only the learning rates are searched from $\eta \in \{1\text{E-}3, 2\text{E-}3, 5\text{E-}3, 1\text{E-}2, 2\text{E-}2, 5\text{E-}2\}$. For VeRA-based methods, following VeRA (Kopiczko et al., 2024), we set the rank $r = 256$. We adopt the AdamW optimizer (Loshchilov & Hutter, 2017), introduce separate learning rates for the classification head and the adapted layers, and determine the learning rates through hyperparameter tuning. Other settings are the same as for LoRA-based methods. The learning rates used for each method are shown in Tables 6, 7, and 8 in Appendix.

Table 1: Performance of different methods on the GLUE benchmark. MNLI-m denotes MNLI with matched test sets, and MNLI-mm denotes MNLI with mismatched test sets. For all tasks, we report accuracy evaluated across 3 runs with mean and standard deviation.

|  | Method | MNLI-m | MNLI-mm | SST2 | QNLI | QQP | RTE | Avg. |
|---|---|---|---|---|---|---|---|---|
| LoRA | LoRA | $88.71_{\pm0.09}$ | $88.19_{\pm0.02}$ | $95.16_{\pm0.09}$ | $91.16_{\pm0.72}$ | $85.33_{\pm1.33}$ | $87.49_{\pm0.15}$ | 89.33 |
|  | FFA-LoRA | $88.83_{\pm0.02}$ | $88.27_{\pm0.03}$ | $94.95_{\pm0.04}$ | $91.52_{\pm0.59}$ | $86.71_{\pm0.07}$ | $86.08_{\pm1.16}$ | 89.39 |
|  | FedDPA-LoRA | $88.99_{\pm0.06}$ | $88.43_{\pm0.05}$ | $95.50_{\pm0.06}$ | $90.74_{\pm1.38}$ | $85.73_{\pm1.73}$ | $87.44_{\pm0.13}$ | 89.47 |
|  | FedSA-LoRA | $\mathbf{90.18}_{\pm0.02}$ | $\mathbf{88.88}_{\pm0.02}$ | $\mathbf{96.00}_{\pm0.04}$ | $\mathbf{92.13}_{\pm0.24}$ | $\mathbf{87.48}_{\pm0.22}$ | $\mathbf{87.93}_{\pm0.11}$ | $\mathbf{90.43}$ |
| rsLoRA | rsLoRA | $88.91_{\pm0.15}$ | $88.33_{\pm0.04}$ | $95.02_{\pm0.24}$ | $91.21_{\pm0.39}$ | $86.73_{\pm0.98}$ | $85.99_{\pm0.34}$ | 89.36 |
|  | FFA-rsLoRA | $89.21_{\pm0.11}$ | $88.45_{\pm0.08}$ | $95.42_{\pm0.17}$ | $91.42_{\pm0.44}$ | $86.93_{\pm1.18}$ | $85.24_{\pm0.21}$ | 89.44 |
|  | FedDPA-rsLoRA | $89.34_{\pm0.11}$ | $88.53_{\pm0.05}$ | $95.56_{\pm0.21}$ | $90.97_{\pm0.76}$ | $86.81_{\pm0.53}$ | $86.26_{\pm0.11}$ | 89.57 |
|  | FedSA-rsLoRA | $\mathbf{90.35}_{\pm0.11}$ | $\mathbf{89.02}_{\pm0.03}$ | $\mathbf{95.78}_{\pm0.08}$ | $\mathbf{92.03}_{\pm0.22}$ | $\mathbf{87.97}_{\pm0.16}$ | $\mathbf{88.00}_{\pm0.10}$ | $\mathbf{90.52}$ |
| VeRA | VeRA | $85.54_{\pm0.10}$ | $85.09_{\pm0.07}$ | $93.53_{\pm0.13}$ | $91.90_{\pm0.17}$ | $82.07_{\pm0.35}$ | $86.31_{\pm0.12}$ | 87.40 |
|  | FFA-VeRA | $86.63_{\pm0.13}$ | $86.22_{\pm0.08}$ | $93.44_{\pm0.05}$ | $92.05_{\pm0.23}$ | $82.23_{\pm0.07}$ | $83.54_{\pm0.59}$ | 87.35 |
|  | FedDPA-VeRA | $86.74_{\pm0.11}$ | $86.35_{\pm0.07}$ | $93.61_{\pm0.32}$ | $90.73_{\pm0.54}$ | $82.11_{\pm0.41}$ | $86.12_{\pm0.12}$ | 87.61 |
|  | FedSA-VeRA | $\mathbf{87.21}_{\pm0.10}$ | $\mathbf{86.52}_{\pm0.04}$ | $\mathbf{93.68}_{\pm0.07}$ | $\mathbf{92.91}_{\pm0.09}$ | $\mathbf{82.56}_{\pm0.05}$ | $\mathbf{87.83}_{\pm0.09}$ | $\mathbf{88.45}$ |

The experimental results are shown in Table 1. From this table, we can observe that the proposed FedSA-LoRA, FedSA-rsLoRA, and FedSA-VeRA consistently outperform other methods across all tasks, demonstrating the effectiveness of the proposed method.

**System Efficiency.** We further compare the proposed method with baselines in terms of system efficiency, following Qu et al. (2022); Zhang et al. (2024a); Lai et al. (2022). The system efficiency in FL consists of communication cost and computation cost. To provide a comprehensive comparison, we detail the number of trainable parameters, the number of communication model parameters per FL round, the computation cost per FL round, and the number of communication rounds needed to reach the predefined target performance on the RTE and QNLI tasks in Table 2. The target performance is defined as 95% of the prediction accuracy provided in LoRA (Hu et al., 2022). Specifically, we define the target performance of the RTE and QNLI tasks as 80.94% and 90.06%, respectively.

Communication cost is a critical factor in FL, as it significantly impacts the overall system efficiency. The communication cost can be roughly estimated by considering the number of transmitted messages required to achieve a target performance, calculated as *# transmitted messages= # communication round × # communicated model parameter*. As shown in Table 2, our FedSA-LoRA requires the smallest communication cost to reach the target performance (with smallest communication rounds and per-round communication parameters). Additionally, while our FedSA-LoRA requires more trainable model parameters and incurs slightly more computation cost per FL round than the baseline FFA-LoRA (22s compared to 20s on RTE task), it is important to note that our model reaches the target performance with fewer communication rounds (#91 compared to #229 on RTE task). These computation and communication costs demonstrate the overall efficiency of our model.

Table 2: Time and space costs for each method on the RTE and QNLI tasks. # Communication round denotes the number of communication rounds to reach the predefined target performance.

| | # Trainable Parm. | # Per-round Communicated Parm. | # Per-round Computation Cost | | # Communication Round | |
|---|---|---|---|---|---|---|
| | | | RTE | QNLI | RTE | QNLI |
| LoRA | 1.83M | 0.78M | 22s | 35s | 167 | 397 |
| FFA-LoRA | 1.44M | 0.39M | 20s | 33s | 229 | 374 |
| FedDPA-LoRA | 2.62M | 0.78M | 23s | 37s | 128 | 325 |
| FedSA-LoRA | 1.83M | 0.39M | 22s | 34s | 91 | 224 |

## 5.2 IN-DEPTH ANALYSES

In this section, we utilize LoRA-based methods to perform in-depth analyses on the natural language understanding tasks of QNLI, SST2, and MNLI-m to assess the impact of factors such as data heterogeneity, the number of clients, and LoRA rank on model performance.

### 5.2.1 EFFECT OF DATA HETEROGENEITY

To investigate the effect of data heterogeneity on model performance, we model an IID partition (Split-1) and two non-IID partitions with Dir (1) and Dir (0.5). The latter two non-IID partitions are referred to as moderate non-IID (Split-2) and severe non-IID (Split-3). The training settings are the same as in Section 5.1.

Table 3: Performance comparison on the QNLI, SST2, and MNLI-m tasks with various degrees of data heterogeneity.

| Method | QNLI | | | SST2 | | | MNLI-m | | |
|---|---|---|---|---|---|---|---|---|---|
| | Split-1 | Split-2 | Split-3 | Split-1 | Split-2 | Split-3 | Split-1 | Split-2 | Split-3 |
| LoRA | 92.92 | 92.44 | 90.60 | 95.30 | 95.53 | 95.26 | 88.52 | 88.35 | 88.80 |
| FFA-LoRA | 92.68 | 92.29 | 91.72 | 95.87 | 95.47 | 94.91 | 88.15 | 88.03 | 88.83 |
| FedSA-LoRA | 92.95 | 93.32 | 92.00 | 96.10 | 96.24 | 95.92 | 89.57 | 89.71 | 90.20 |

The results are provided in Table 3. From these results, we can observe that the proposed FedSA-LoRA consistently outperforms other baselines, demonstrating its adaptability and robustness in various heterogeneous data scenarios. Additionally, as data heterogeneity increases, the improvement of the proposed method also increases. Specifically, FedSA-LoRA improves accuracy by 0.03%, 0.88%, and 1.84% on the QNLI task from IID to severe non-IID compared with LoRA, and by 1.05%, 1.36%, and 1.4% on the MNLI-m task. This indicates that the proposed method is more effective when non-IID conditions are more severe. This phenomenon is consistent with Figure 2, which shows that with increased data heterogeneity, the similarity of $B$ matrices between different clients decreases. Therefore, when the non-IID conditions are more severe, the advantages of keeping $B$ locally become more pronounced. In this case, the learned $B$ will be less similar, highlighting the need for personalization.

### 5.2.2 EFFECT OF NUMBER OF CLIENTS

In Section 5.1, we demonstrated the effectiveness of the proposed method on a small number of clients, i.e., three clients. In this section, we show the superiority of FedSA-LoRA compared to other baselines on a larger number of clients, i.e., from 10 to 100 clients. Specifically, we use the same non-IID split, i.e., Dir (0.5), to divide the data into 10, 20, and 100 clients. The training settings are the same as in Section 5.1 and the results are shown in Table 4.

It can be concluded that FedSA-LoRA not only outperforms other methods with a small number of clients (i.e., 3 clients) but also shows superior performance with a large number of clients (i.e., from 10 to 100 clients), demonstrating the adaptability and robustness of the proposed FedSA-LoRA across various client numbers.

Table 4: Performance comparison on the QNLI, SST2, and MNLI-m tasks with different number of clients. We apply full participation for FL system with 10 and 20 clients, and apply client sampling with rate 0.3 for FL system with 100 clients.

| Method | QNLI | | | SST2 | | | MNLI-m | | |
|---|---|---|---|---|---|---|---|---|---|
| | 10 clients | 20 clients | 100 clients | 10 clients | 20 clients | 100 clients | 10 clients | 20 clients | 100 clients |
| LoRA | 91.32 | 91.23 | 90.32 | 96.68 | 93.16 | 96.68 | 86.94 | 88.50 | 88.13 |
| FFA-LoRA | 91.47 | 91.70 | 91.27 | 96.59 | 93.31 | 96.33 | 86.76 | 88.60 | 87.86 |
| FedSA-LoRA | 91.97 | 92.54 | 91.48 | 96.83 | 94.21 | 97.02 | 88.59 | 89.05 | 88.82 |

### 5.2.3 EFFECT OF LoRA RANK

The adapter parameter budget (i.e., rank $r$) is a key factor in LoRA performance. In this section, we experiment with rank $r \in \{2, 4, 8, 16\}$ on the QNLI, SST2, and MNLI-m tasks to test its influence on model performance, keeping other settings unchanged compared to Section 5.1.

Table 5: Performance comparison on the QNLI, SST2, and MNLI-m tasks with different LoRA ranks $r$.

| Rank | Method | QNLI | SST2 | MNLI-m | Rank | Method | QNLI | SST2 | MNLI-m |
|---|---|---|---|---|---|---|---|---|---|
| | LoRA | 92.07 | 94.69 | 88.29 | | LoRA | 92.71 | 94.20 | 88.43 |
| $r = 2$ | FFA-LoRA | 91.02 | 93.95 | 87.98 | $r = 4$ | FFA-LoRA | 92.97 | 94.61 | 88.24 |
| | FedSA-LoRA | 92.69 | 95.67 | 89.14 | | FedSA-LoRA | 93.10 | 95.69 | 88.98 |
| | LoRA | 90.69 | 95.26 | 88.80 | | LoRA | 90.59 | 94.98 | 88.78 |
| $r = 8$ | FFA-LoRA | 91.72 | 94.91 | 88.83 | $r = 16$ | FFA-LoRA | 91.62 | 94.13 | 89.25 |
| | FedSA-LoRA | 92.00 | 95.92 | 90.20 | | FedSA-LoRA | 92.03 | 95.78 | 89.59 |

The results, as shown in Table 5, demonstrate that the proposed FedSA-LoRA outperforms other methods across various LoRA rank values, showcasing the adaptability and robustness of FedSA-LoRA in different scenarios.

## 5.3 NATURAL LANGUAGE GENERATION

For the natural language generation tasks, we adopt the pre-trained LLaMA3-8B (Meta, 2024) from the HuggingFace Transformers library (Wolf et al., 2020), using the GSM8K dataset (Cobbe et al., 2021) and the CodeSearchNet dataset (Husain et al., 2019) for evaluation. For the GSM8K dataset (Cobbe et al., 2021), Following (Kuang et al., 2023), we split the data into three clients under an IID distribution, and other optimization hyperparameters are the same as in that work. The results of LoRA, FFA-LoRA, and FedSA-LoRA are 46.23, 46.32, and 46.63, respectively, and the generated examples are shown in Table 9 in Appendix. From the given example, it can be seen that both LoRA and FFA-LoRA have reasoning errors, but FedSA-LoRA can reason accurately, demonstrating the superiority of the proposed method in complex natural language generation tasks.

For an additional dataset on the code generation task. We choose the CodeSearchNet dataset (Husain et al., 2019) and use the default non-IID partitioning provided in (Kuang et al., 2023). The performance scores for LoRA, FFA-LoRA, and our FedSA-LoRA are 58.34, 58.57, and 59.66, respectively, which further validates the effectiveness of our method in generation tasks.

## 6 CONCLUSION

In this work, we discover that when combining LoRA with FL, $A$ matrices are responsible for learning general knowledge, while $B$ matrices focus on capturing client-specific knowledge. Building upon this finding, we introduce Federated Share-A Low-Rank Adaptation (FedSA-LoRA), which employs two low-rank trainable matrices $A$ and $B$ to model the weight update, but only $A$ matrices are shared with the server for aggregation. By sharing the $A$ matrices that learn general knowledge with the server for aggregation, while keeping the $B$ matrices that model client-specific knowledge locally, the learning abilities of LoRA combined with FL can be enhanced. Moreover, we explore the relationship between the learned $A$ and $B$ matrices in other LoRA variants, such as rsLoRA and VeRA, revealing a consistent pattern. Consequently, we extend our FedSA-LoRA method to these LoRA variants, resulting in FedSA-rsLoRA and FedSA-VeRA. By doing so, we establish a general paradigm for integrating LoRA with FL, offering guidance for future work on subsequent LoRA variants combined with FL.

## ACKNOWLEDGMENTS

This work was supported by National Natural Science Foundation of China (62306253, U24A201397), Early career fund (27204623), and Guangdong Natural Science Fund-General Programme (2024A1515010233).

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

# A APPENDIX

## A.1 PROOF OF LEMMA 1

*Proof.* Consider fine-tuning $B$ while freezing $A = Q$. The loss function in Eq. (3) becomes:

$$\mathcal{L} = \mathbb{E}_{(x_t, y_t)}[\|y_t - (W_0 + BQ)x_t\|_2^2]. \tag{7}$$

Then, the gradient of Eq. (7) w.r.t. $B$ is:

$$
\begin{aligned}
\frac{\partial \mathcal{L}}{\partial B} &= \frac{\partial \mathbb{E}_{(x_t, y_t)}[\|y_t - (W_0 + BQ)x_t\|_2^2]}{\partial B} \\
&= \frac{\partial \mathbb{E}[\|W_t x_t - (W_0 + BQ)x_t\|_2^2]}{\partial B} \\
&= \frac{\partial \mathbb{E}[\|(W_0 + \Delta_W)x_t - (W_0 + BQ)x_t\|_2^2]}{\partial B} \\
&= \frac{\partial \mathbb{E}[\|(\Delta_W - BQ)x_t\|_2^2]}{\partial B} \\
&= \mathbb{E}[2[(\Delta_W - BQ)x_t](-x_t^T Q^T)] \\
&= \mathbb{E}[2(BQ - \Delta_W)x_t x_t^T Q^T].
\end{aligned} \tag{8}
$$

To obtain the optimal $B^*$, we set Eq. (8) to zero, which means:

$$
\begin{aligned}
\mathbb{E}[2(BQ - \Delta_W)x_t x_t^T Q^T] &= 0 \\
2BQ\mathbb{E}[x_t x_t^T]Q^T - 2\Delta_W \mathbb{E}[x_t x_t^T]Q^T &= 0 \\
2BQ\mathbb{E}[x_t x_t^T]Q^T - 2\Delta_W \mathbb{E}[x_t x_t^T]Q^T &= 0 \\
BQ\mathbb{E}[x_t x_t^T]Q^T &= \Delta_W \mathbb{E}[x_t x_t^T]Q^T \\
B &= \Delta_W \mathbb{E}[x_t x_t^T]Q^T (Q\mathbb{E}[x_t x_t^T]Q^T)^{-1}.
\end{aligned} \tag{9}
$$

Therefore, we obtain $B^* = \Delta_W \mathbb{E}[x_t x_t^T]Q^T (Q\mathbb{E}[x_t x_t^T]Q^T)^{-1}$.

When fine-tuning $A$ with fixed $B = U$. The loss function in Eq. (3) becomes:

$$\mathcal{L} = \mathbb{E}_{(x_t, y_t)}[\|y_t - (W_0 + UA)x_t\|_2^2]. \tag{10}$$

Then, the gradients of Eq. (7) w.r.t. $A$ is:

$$
\begin{aligned}
\frac{\partial \mathcal{L}}{\partial A} &= \frac{\partial \mathbb{E}_{(x_t, y_t)}[\|y_t - (W_0 + UA)x_t\|_2^2]}{\partial A} \\
&= \frac{\partial \mathbb{E}[\|W_t x_t - (W_0 + UA)x_t\|_2^2]}{\partial A} \\
&= \frac{\partial \mathbb{E}[\|(W_0 + \Delta_W)x_t - (W_0 + BQ)x_t\|_2^2]}{\partial A} \\
&= \frac{\partial \mathbb{E}[\|(\Delta_W - UA)x_t\|_2^2]}{\partial A} \\
&= \mathbb{E}[2U^T[(\Delta_W - UA)x_t]x_t^T]
\end{aligned} \tag{11}
$$

To obtain the optimal $A^*$, we set Eq. (11) to zero, which means:

$$
\begin{aligned}
\mathbb{E}[2U^T[(\Delta_W - UA)x_t]x_t^T] &= 0 \\
2U^T \Delta_W \mathbb{E}[x_t x_t^T] - 2U^T UA\mathbb{E}[x_t x_t^T] &= 0 \\
U^T UA\mathbb{E}[x_t x_t^T] &= U^T \Delta_W \mathbb{E}[x_t x_t^T] \\
A &= U^{-1}\Delta_W.
\end{aligned} \tag{12}
$$

Thus, we obtain $A^* = U^{-1}\Delta_W$. $\qquad\square$

### A.2 PROOF OF THEOREM 1

*Proof.* Let $W_i^{(t)} = W_0 + B_i^{(t)} A_i^{(t)}$ be the model parameters maintained in the $i$-th client at the $t$-th step. Let $\mathcal{I}_E$ be the set of global synchronization steps, i.e., $\mathcal{I}_E = \{nE \mid n = 1, 2, \cdots\}$. If $t + 1 \in \mathcal{I}_E$, which represents the time step for communication, then the one-step update of the proposed method for the $i$-th client can be described as follows:

$$\begin{pmatrix} B_i^{(t)} \\ A_i^{(t)} \end{pmatrix} \xrightarrow[\text{update of } B_i^{(t)} \text{ and } A_i^{(t)}]{} \begin{pmatrix} B_i^{(t+1)} \\ A_i^{(t+1)} \end{pmatrix} \xrightarrow[\text{if } t+1 \in \mathcal{I}_E]{} \begin{pmatrix} B_i^{(t+1)} \\ \frac{1}{m} \sum_{j=1}^{m} A_j^{(t+1)} \end{pmatrix}.$$

For convenience, we denote the parameters in each sub-step above as follows:

$$W_i^{(t)} = W_0 + B_i^{(t)} A_i^{(t)},$$
$$U_i^{(t)} = W_0 + B_i^{(t+1)} A_i^{(t+1)},$$
$$V_i^{(t)} = W_0 + B_i^{(t+1)} \frac{1}{m} \sum_{j=1}^{m} A_j^{(t+1)},$$
$$W_i^{(t+1)} = \begin{cases} U_i^{(t)} & \text{if } t + 1 \notin \mathcal{I}_E, \\ V_i^{(t)} & \text{if } t + 1 \in \mathcal{I}_E. \end{cases}$$

Here, the variable $U_i^{(t)}$ represents the immediate result of one sub-step update from the parameter of the previous sub-step $W_i^{(t)}$, and $V_i^{(t)}$ represents the parameter obtained after communication steps (if applicable). Furthermore, we denote the learning rate for the $i$-th client at the $t$-th step as $\eta_{i,t}$, and the stochastic gradient at step $t$ as follows:

$$\nabla_B \mathcal{L}_i(W_i^{(t)}, \xi_{i,t}) = \nabla_W \mathcal{L}_i(W_i^{(t)}, \xi_{i,t}) A_i^{(t)\top},$$
$$\nabla_A \mathcal{L}_i(W_i^{(t)}, \xi_{i,t}) = B_i^{(t)\top} \nabla_W \mathcal{L}_i(W_i^{(t)}, \xi_{i,t}),$$
$$\nabla_B \mathcal{L}_i(W_i^{(t)}) = \mathbb{E}[\nabla_B \mathcal{L}_i(W_i^{(t)}, \xi_{i,t})],$$
$$\nabla_A \mathcal{L}_i(W_i^{(t)}) = \mathbb{E}[\nabla_A \mathcal{L}_i(W_i^{(t)}, \xi_{i,t})],$$

where $\xi_{i,t}$ is the data uniformly chosen from the local data set of client $i$ at step $t$.

Next, we apply the inequality from the smoothness Assumption 1 to each sub-step of the one-step update for client $i$. Firstly, by the smoothness of $\mathcal{L}_i$, we have:

$$\mathcal{L}_i(U_i^{(t)}) \leq \mathcal{L}_i(W_i^{(t)}) + \langle U_i^{(t)} - W_i^{(t)}, \nabla_W \mathcal{L}_i(W_i^{(t)}) \rangle_F + \frac{L}{2} \left\| U_i^{(t)} - W_i^{(t)} \right\|_F^2. \quad (13)$$

Since
$$B_i^{(t+1)} = B_i^{(t)} - \eta_{i,t} \nabla_B \mathcal{L}_i(W_i^{(t)}, \xi_{i,t})$$
$$= B_i^{(t)} - \eta_{i,t} \nabla_W \mathcal{L}_i(W_i^{(t)}, \xi_{i,t}) A_i^{(t)\top},$$

and
$$A_i^{(t+1)} = A_i^{(t)} - \eta_{i,t} \nabla_A \mathcal{L}_i(W_i^{(t)}, \xi_{i,t})$$
$$= A_i^{(t)} - \eta_{i,t} B_i^{(t)\top} \nabla_W \mathcal{L}_i(W_i^{(t)}, \xi_{i,t}),$$

we have:

$$U_i^{(t)} - W_i^{(t)}$$
$$= B_i^{(t+1)} A_i^{(t+1)} - B_i^{(t)} A_i^{(t)}$$
$$= \left( B_i^{(t)} - \eta_{i,t} \nabla_W \mathcal{L}_i(W_i^{(t)}, \xi_{i,t}) A_i^{(t)\top} \right) \left( A_i^{(t)} - \eta_{i,t} B_i^{(t)\top} \nabla_W \mathcal{L}_i(W_i^{(t)}, \xi_{i,t}) \right) - B_i^{(t)} A_i^{(t)}$$
$$= \eta_{i,t}^2 \nabla_W \mathcal{L}_i(W_i^{(t)}, \xi_{i,t}) A_i^{(t)\top} B_i^{(t)\top} \nabla_W \mathcal{L}_i(W_i^{(t)}, \xi_{i,t}) - \eta_{i,t} \nabla_W \mathcal{L}_i(W_i^{(t)}, \xi_{i,t}) A_i^{(t)\top} A_i^{(t)}$$
$$\quad - \eta_{i,t} B_i^{(t)} B_i^{(t)\top} \nabla_W \mathcal{L}_i(W_i^{(t)}, \xi_{i,t}).$$

Then, for the second term on the right side of Eq. (13), according to the law of total expectation, we have:

$$
\mathbb{E}\left[\left\langle U_i^{(t)} - W_i^{(t)}, \nabla_W \mathcal{L}_i(W_i^{(t)})\right\rangle_F\right]
$$

$$
= \eta_{i,t}^2 \mathbb{E}\left[\left\langle \nabla_W \mathcal{L}_i(W_i^{(t)}, \xi_{i,t}) A_i^{(t)\top} B_i^{(t)\top} \nabla_W \mathcal{L}_i(W_i^{(t)}, \xi_{i,t}), \nabla_W \mathcal{L}_i(W_i^{(t)})\right\rangle_F\right]
$$

$$
- \eta_{i,t} \mathbb{E}\left[\left\langle \nabla_W \mathcal{L}_i(W_i^{(t)}, \xi_{i,t}) A_i^{(t)\top} A_i^{(t)}, \nabla_W \mathcal{L}_i(W_i^{(t)})\right\rangle_F\right]
$$

$$
- \eta_{i,t} \mathbb{E}\left[\left\langle B_i^{(t)} B_i^{(t)\top} \nabla_W \mathcal{L}_i(W_i^{(t)}, \xi_{i,t}), \nabla_W \mathcal{L}_i(W_i^{(t)})\right\rangle_F\right]
$$

$$
= \eta_{i,t}^2 \left\langle \nabla_W \mathcal{L}_i(W_i^{(t)}) A_i^{(t)\top} B_i^{(t)\top} \nabla_W \mathcal{L}_i(W_i^{(t)}), \nabla_W \mathcal{L}_i(W_i^{(t)})\right\rangle_F
$$

$$
- \eta_{i,t} \left\langle \nabla_W \mathcal{L}_i(W_i^{(t)}) A_i^{(t)\top} A_i^{(t)}, \nabla_W \mathcal{L}_i(W_i^{(t)})\right\rangle_F
$$

$$
- \eta_{i,t} \left\langle B_i^{(t)} B_i^{(t)\top} \nabla_W \mathcal{L}_i(W_i^{(t)}), \nabla_W \mathcal{L}_i(W_i^{(t)})\right\rangle_F.
$$

Since

$$
\left\langle \nabla_W \mathcal{L}_i(W_i^{(t)}) A_i^{(t)\top} B_i^{(t)\top} \nabla_W \mathcal{L}_i(W_i^{(t)}), \nabla_W \mathcal{L}_i(W_i^{(t)})\right\rangle_F
$$

$$
\leq \left\| \nabla_W \mathcal{L}_i(W_i^{(t)}) A_i^{(t)\top} B_i^{(t)\top} \nabla_W \mathcal{L}_i(W_i^{(t)})\right\| \left\| \nabla_W \mathcal{L}_i(W_i^{(t)})\right\|_F
$$

$$
\leq \left\| A_i^{(t)}\right\| \left\| B_i^{(t)}\right\|_F \left\| \nabla_W \mathcal{L}_i(W_i^{(t)})\right\|_F^3
$$

$$
\leq C_A C_B G^3,
$$

and if we assume there exists $c_A > 0$ such that $\forall t$:

$$
\left\langle A_i^{(t)\top} A_i^{(t)}, \nabla_W \mathcal{L}_i(W_i^{(t)})^\top \nabla_W \mathcal{L}_i(W_i^{(t)})\right\rangle_F \geq c_A \left\| \nabla_W \mathcal{L}_i(W_i^{(t)})\right\|_F^2,
$$

then we have:

$$
\left\langle \nabla_W \mathcal{L}_i(W_i^{(t)}) A_i^{(t)\top} A_i^{(t)}, \nabla_W \mathcal{L}_i(W_i^{(t)})\right\rangle_F = \mathrm{Tr}\left[\left(\nabla_W \mathcal{L}_i(W_i^{(t)}) A_i^{(t)\top} A_i^{(t)}\right)^\top \nabla_W \mathcal{L}_i(W_i^{(t)})\right]
$$

$$
= \mathrm{Tr}\left[A_i^{(t)\top} A_i^{(t)} \nabla_W \mathcal{L}_i(W_i^{(t)})^\top \nabla_W \mathcal{L}_i(W_i^{(t)})\right]
$$

$$
= \left\langle A_i^{(t)\top} A_i^{(t)}, \nabla_W \mathcal{L}_i(W_i^{(t)})^\top \nabla_W \mathcal{L}_i(W_i^{(t)})\right\rangle_F
$$

$$
\geq c_A \left\| \nabla_W \mathcal{L}_i(W_i^{(t)})\right\|_F^2.
$$

And similarly if we assume there exists $c_B > 0$ such that $\forall t$:

$$
\left\langle B_i^{(t)\top} B_i^{(t)}, \nabla_W \mathcal{L}_i(W_i^{(t)})^\top \nabla_W \mathcal{L}_i(W_i^{(t)})\right\rangle_F \geq c_B \left\| \nabla_W \mathcal{L}_i(W_i^{(t)})\right\|_F^2,
$$

then we have:

$$
\left\langle B_i^{(t)} B_i^{(t)\top} \nabla_W \mathcal{L}_i(W_i^{(t)}), \nabla_W \mathcal{L}_i(W_i^{(t)})\right\rangle_F = \mathrm{Tr}\left[\left(B_i^{(t)} B_i^{(t)\top} \nabla_W \mathcal{L}_i(W_i^{(t)})\right)^\top \nabla_W \mathcal{L}_i(W_i^{(t)})\right]
$$

$$
= \mathrm{Tr}\left[\nabla_W \mathcal{L}_i(W_i^{(t)})^\top B_i^{(t)} B_i^{(t)\top} \nabla_W \mathcal{L}_i(W_i^{(t)})\right]
$$

$$
= \mathrm{Tr}\left[\nabla_W \mathcal{L}_i(W_i^{(t)}) \nabla_W \mathcal{L}_i(W_i^{(t)})^\top B_i^{(t)} B_i^{(t)\top}\right]
$$

$$
= \left\langle \nabla_W \mathcal{L}_i(W_i^{(t)}) \nabla_W \mathcal{L}_i(W_i^{(t)})^\top, B_i^{(t)} B_i^{(t)\top}\right\rangle_F
$$

$$
\geq c_B \left\| \nabla_W \mathcal{L}_i(W_i^{(t)})\right\|_F^2, \tag{14}
$$

where we use the cyclic property of the trace for the third equality above. We further get:

$$\mathbb{E}\left[\left\langle U_i^{(t)} - W_i^{(t)}, \nabla_W \mathcal{L}_i(W_i^{(t)})\right\rangle_F\right]$$

$$\leq \eta_{i,t}^2 C_A C_B G^3 - \eta_{i,t} c_A^2 \left\|\nabla_W \mathcal{L}_i(W_i^{(t)})\right\|_F^2 - \eta_{i,t} c_B^2 \left\|\nabla_W \mathcal{L}_i(W_i^{(t)})\right\|_F^2. \tag{15}$$

Similarly, we know:

$$\mathbb{E}[\|U_i^{(t)} - W_i^{(t)}\|_F^2] = \mathbb{E}[\|\eta_{i,t}^2 \nabla_W \mathcal{L}_i(W_i^{(t)}, \xi_{i,t}) A_i^{(t)\top} B_i^{(t)\top} \nabla_W \mathcal{L}_i(W_i^{(t)}, \xi_{i,t})$$

$$- \eta_{i,t} \nabla_W \mathcal{L}_i(W_i^{(t)}, \xi_{i,t}) A_i^{(t)\top} A_i^{(t)} - \eta_{i,t} B_i^{(t)} B_i^{(t)\top} \nabla_W \mathcal{L}_i(W_i^{(t)}, \xi_{i,t})\|_F^2]$$

$$\leq 3\eta_{i,t}^4 C_A^2 C_B^2 G^4 + 3\eta_{i,t}^2 C_A^2 G^2 + 3\eta_{i,t}^2 C_B^2 G^2. \tag{16}$$

Plugging Eq. (14), Eq. (15), and Eq. (16) into Eq. (13), we have:

$$\mathcal{L}_i\left(U_i^{(t)}\right) \leq \mathcal{L}_i\left(W_i^{(t)}\right) + \eta_{i,t}^2 C_A C_B G^3 - \eta_{i,t} c_A \left\|\nabla_W \mathcal{L}_i(W_i^{(t)})\right\|_F^2 - \eta_{i,t} c_B \left\|\nabla_W \mathcal{L}_i(W_i^{(t)})\right\|_F^2$$

$$+ \frac{3}{2}\eta_{i,t}^4 C_A^2 C_B^2 G^4 L + \frac{3}{2}\eta_{i,t}^2 C_A^2 G^2 L + \frac{3}{2}\eta_{i,t}^2 C_B^2 G^2 L. \tag{17}$$

Secondly, by the smoothness of $\mathcal{L}_i$, we have:

$$\mathcal{L}_i\left(V_i^{(t)}\right) \leq \mathcal{L}_i\left((U_i^{(t)})\right) + \left\langle V_i^{(t)} - U_i^{(t)}, \nabla_W \mathcal{L}_i(U_i^{(t)})\right\rangle_F + \frac{L}{2}\left\|V_i^{(t)} - U_i^{(t)}\right\|_F^2. \tag{18}$$

Since

$$V_i^{(t)} - U_i^{(t)} = B_i^{(t+1)} A_i^{(t+1)} - \frac{1}{m} B_i^{(t+1)} \sum_{j=1}^m A_j^{(t+1)}$$

$$= B_i^{(t+1)} \frac{1}{m} \sum_{j=1}^m \left(A_i^{(t+1)} - A_j^{(t+1)}\right),$$

and

$$A_j^{(t+1)} = A_j^{(t-E+1)} - \sum_{t_0=t-E+1}^t \eta_{j,t_0} \nabla_A \mathcal{L}_j(W_j^{(t_0)}; \xi_{j,t_0})$$

$$= A_j^{(t-E+1)} - \sum_{t_0=t-E+1}^t \eta_{j,t_0} B_j^{(t_0)\top} \nabla_W \mathcal{L}_j(W_j^{(t_0)}; \xi_{j,t_0}),$$

we know:

$$V_i^{(t)} - U_i^{(t)} = B_i^{(t+1)}\left(-\frac{1}{m}\sum_{j=1}^m \sum_{t_0=t-E+1}^t \eta_{j,t_0} B_j^{(t_0)\top}\left(\nabla \mathcal{L}_i(W_i^{(t_0)}; \xi_{i,t_0}) - \nabla \mathcal{L}_j(W_j^{(t_0)}; \xi_{j,t_0})\right)\right).$$

Therefore,

$$\mathbb{E}\left[\left\|V_i^{(t)} - U_i^{(t)}\right\|_F^2\right]$$

$$= \mathbb{E}\left[\left\|B_i^{(t+1)}\right\|_F^2\right\| - \frac{1}{m}\sum_{j=1}^m \sum_{t_0=t-E+1}^t \eta_{j,t_0} B_j^{(t_0)\top}\left(\nabla \mathcal{L}_i(W_i^{(t_0)}; \xi_{i,t_0}) - \nabla \mathcal{L}_j(W_j^{(t_0)}; \xi_{j,t_0})\right)\right\|_F^2\right]$$

$$\leq C_B^2 \frac{E}{m}\sum_{j=1}^m \sum_{t_0=t-E+1}^t \eta_{j,t_0}^2 \mathbb{E}\left[\left\|B_j^{(t_0)\top}\left(\nabla \mathcal{L}_i(W_i^{(t_0)}; \xi_{i,t_0}) - \nabla \mathcal{L}_j(W_j^{(t_0)}; \xi_{j,t_0})\right)\right\|_F^2\right]$$

$$\leq \frac{C_B^4 E}{m}\sum_{j=1}^m \sum_{t_0=t-E+1}^t \eta_{j,t_0}^2 \mathbb{E}\left[\left\|\nabla \mathcal{L}_i(W_i^{(t_0)}; \xi_{i,t_0}) - \nabla \mathcal{L}_j(W_j^{(t_0)}; \xi_{j,t_0})\right\|_F^2\right]$$

$$\leq \frac{4 C_B^4 E G^2}{m}\sum_{j=1}^m \sum_{t_0=t-E+1}^t \eta_{j,t_0}^2, \tag{19}$$

where we use Assumption 2 to derive that:

$$\mathbb{E}\left[\left\|\nabla\mathcal{L}_i(W_i^{(t_0)};\xi_{i,t_0}) - \nabla\mathcal{L}_j(W_j^{(t_0)};\xi_{j,t_0})\right\|_F^2\right]$$

$$\leq 2\mathbb{E}\left[\left\|\nabla\mathcal{L}_i(W_i^{(t_0)};\xi_{i,t_0})\right\|_F^2\right] + 2\mathbb{E}\left[\left\|\nabla\mathcal{L}_i(W_j^{(t_0)};\xi_{j,t_0})\right\|_F^2\right]$$

$$\leq 4G^2.$$

Furthermore,

$$\left\langle V_i^{(t)} - U_i^{(t)}, \nabla_W L_i(U_i^{(t)})\right\rangle \leq \frac{1}{2\eta_{i,t}}\left\|V_i^{(t)} - U_i^{(t)}\right\|_F^2 + \frac{1}{2}\eta_{i,t}\left\|\nabla_W\mathcal{L}_i(U_i^{(t)})\right\|_F^2$$

$$\leq \frac{2C_B^4 EG^2}{\eta_{i,t}m}\sum_{j=1}^m\sum_{t_0=t-E+1}^t \eta_{j,t_0}^2 + \frac{1}{2}\eta_{i,t}G^2. \tag{20}$$

Plugging Eq. (19) and Eq. (20) into Eq. (18), we have (choose constant learning rate $\eta_{i,t} = \eta$):

$$\mathcal{L}_i\left(V_i^{(t)}\right) \leq \mathcal{L}_i\left(U_i^{(t)}\right) + (2C_B^2 E^2 G^2 + \frac{1}{2}G^2)\eta + 2\eta^2 C_B^4 E^2 G^2 L. \tag{21}$$

Combining Eq. (17) and Eq. (21), we have (choose constant learning rate $\eta_{i,t} = \eta$):

$$\mathcal{L}_i\left(W_i^{(t+1)}\right) \leq \mathcal{L}_i\left((W_i^{(t)}\right) + (2C_B^2 E^2 G^2 + \frac{1}{2}G^2)\eta + \frac{3}{2}\eta^4 C_A^2 C_B^2 G^4 L$$

$$+ (C_A C_B G^2 + \frac{3}{2}C_A^2 LG^2 + \frac{3}{2}C_B^2 LG^2 + 2C_B^4 E^2 LG^2)\eta^2$$

$$- \eta(c_A + c_B)\left\|\nabla_W\mathcal{L}_i(W_i^{(t)})\right\|^2,$$

which is equivalent to:

$$\eta(c_A + c_B)\left\|\nabla_W\mathcal{L}_i(W_i^{(t)})\right\|_F^2 \leq \mathcal{L}_i\left((W_i^{(t)}\right) - \mathcal{L}_i\left(W_i^{(t+1)}\right)$$

$$+ (2C_B^2 E^2 G^2 + \frac{1}{2}G^2)\eta + \frac{3}{2}\eta^4 C_A^2 C_B^2 G^4 L$$

$$+ (C_A C_B G^2 + \frac{3}{2}C_A^2 LG^2 + \frac{3}{2}C_B^2 LG^2 + 2C_B^4 E^2 LG^2)\eta^2.$$

Choosing $M$ which satisfies $(2C_B^2 E^2 G^2 + \frac{1}{2}G^2)\eta + \frac{3}{2}\eta^4 C_A^2 C_B^2 G^4 L + (C_A C_B G^2 + \frac{3}{2}C_A^2 LG^2 + \frac{3}{2}C_B^2 LG^2 + 2C_B^4 E^2 LG^2)\eta^2 \leq M\eta^2$, we get:

$$\left\|\nabla_W\mathcal{L}_i(W_i^{(t)})\right\|_F^2 \leq \frac{\mathcal{L}_i\left((W_i^{(t)}\right) - \mathcal{L}_i\left(W_i^{(t+1)}\right)}{\eta(c_A + c_B)} + \frac{M\eta}{c_A + c_B}. \tag{22}$$

Now, by repeatedly applying Eq. (22) for different values of $t$ and summing up the results, we get:

$$\sum_{t=1}^T\left\|\nabla_W\mathcal{L}_i(W_i^{(t)})\right\|_F^2 \leq \frac{\mathcal{L}_i\left((W_i^{(1)}\right) - \mathcal{L}_i(W_i^*)}{\eta(c_A^2 + c_B^2)} + \eta\frac{M}{c_A + c_B}T. \tag{23}$$

Dividing both side of Eq. (23) by $T$, we get:

$$\frac{1}{T}\sum_{t=1}^T\left\|\nabla_W\mathcal{L}_i(W_i^{(t)})\right\|_F^2 \leq \frac{\mathcal{L}_i\left((W_i^{(1)}\right) - \mathcal{L}_i(W_i^*)}{\eta(c_A + c_B)T} + \eta\frac{M}{c_A + c_B}. \tag{24}$$

Let us assume that $\mathcal{L}_i(W_i^{(1)}) - \mathcal{L}_i(W_i^*) \leq D, \forall i$, and we set $\eta = \sqrt{\frac{D}{MT}}$. Then, we have:

$$\frac{1}{T}\sum_{t=1}^T\left\|\nabla_W\mathcal{L}_i(W_i^{(t)})\right\|_F^2 \leq \frac{2}{c_A + c_B}\sqrt{\frac{DM}{T}}. \tag{25}$$

Thus, we can obtain:

$$\frac{1}{mT}\sum_{i=1}^m\sum_{t=1}^T\left\|\nabla_W\mathcal{L}_i(W_i^{(t)})\right\|_F^2 \leq \frac{2}{c_A + c_B}\sqrt{\frac{DM}{T}}. \tag{26}$$

$\square$

A.3 FURTHER EXPLANATIONS ABOUT AGGREGATION ERRORS.

In this section, we provide more explanation about what aggregation errors are and why people do not achieve the "ideal" model update.

When introducing LoRA into FL, the update for the $i$-the client is given by $\Delta_{W_i} = B_i A_i$. The "ideal" model update for server aggregation should be:

$$\Delta_W^* = \frac{1}{m} \sum_{i=1}^{m} \Delta_{W_i} = \frac{1}{m} \sum_{i=1}^{m} B_i A_i = \frac{1}{m}(B_1 A_1 + B_2 A_2 + \cdots + B_m A_m). \quad (27)$$

However, in practical LoRA training, the matrices $A_i$ and $B_i$, not the original $W_i$, are trainable. Thus, we cannot directly average the $\Delta_{W_i}$; instead, we can only separately average $A_i$ and $B_i$, and then combine them to obtain the update:

$$\begin{aligned} \Delta_W =& (\frac{1}{m} \sum_{i=1}^{m} B_i)(\frac{1}{m} \sum_{i=1}^{m} A_i) \\ =& \frac{1}{m}(B_1 + B_2 + \cdots + B_m)\frac{1}{m}(A_1 + A_2 + \cdots + A_m) \\ =& \frac{1}{m^2}(B_1 A_1 + B_1 A_2 + \cdots + B_1 A_m + B_2 A_1 \\ & + B_2 A_2 + \cdots + B_2 A_m + \cdots + B_m A_1 + B_m A_2 + \cdots + B_m A_m), \end{aligned} \quad (28)$$

which differs from the "ideal" model update in Eq. (27). This difference introduces aggregation errors.

Regarding why the "ideal" model update is not achieved, it is because the matrices $A_i$ and $B_i$ are trainable, not the original $W_i$. While it is possible to first combine $A_i$ and $B_i$ to obtain $\Delta_{W_i}$ and then average them to achieve the "ideal" update, we cannot decompose $W$ back into $A$ and $B$ for further federated training. This limitation prevents people from performing the "ideal" model update.

A.4 HYPERPARAMETERS

Tables 6 and 7 show the learning rates used for LoRA-based methods and rsLoRA-based methods, respectively. For the VeRA-based methods, we first tried using the SGD optimizer (Ruder, 2016) with a search learning rate from $\eta \in \{5E\text{-}3, 1E\text{-}2, 2E\text{-}2, 5E\text{-}2, 1E\text{-}1\}$ as adopted in LoRA-based methods, but we found the performance to be significantly worse than that of LoRA-based methods. For example, the best performance among the three VeRA-based methods (i.e., VeRA, FFA-VeRA, and FedSA-VeRA) is 53.73% on the MNLI-m task, which is significantly worse than that of the LoRA-based methods (90.20%). Thus, we chose the AdamW optimizer (Loshchilov & Hutter, 2017) and introduced separate learning rates for the classification head and the adapted layers as used in VeRA (Kopiczko et al., 2024). The learning rates used for VeRA-based methods are shown in Table 8.

Table 6: The learning rates used for LoRA-based methods on the GLUE benchmark.

| Method | MNLI-m | MNLI-mm | SST2 | QNLI | QQP | RTE |
|---|---|---|---|---|---|---|
| LoRA | 1E-2 | 1E-2 | 2E-2 | 1E-2 | 1E-2 | 1E-2 |
| FFA-LoRA | 5E-2 | 5E-2 | 5E-2 | 2E-2 | 5E-2 | 2E-2 |
| FedDPA-LoRA | 1E-2 | 1E-2 | 1E-2 | 5E-2 | 5E-2 | 1E-2 |
| FedSA-LoRA | 2E-2 | 2E-2 | 1E-2 | 5E-3 | 2E-2 | 1E-2 |

A.5 EXAMPLES OF GENERATED ANSWER FOR GSM8K DATASETS

The results of the GSM8K dataset are shown in Table 9, demonstrating that the proposed FedSA-LoRA outperforms other methods in complex natural language generation tasks. From the given example, it can be seen that both LoRA and FFA-LoRA have reasoning errors, but FedSA-LoRA can reason accurately, demonstrating the superiority of the proposed method.

Table 7: The learning rates used for rsLoRA-based methods on the GLUE benchmark.

| Method | MNLI-m | MNLI-mm | SST2 | QNLI | QQP | RTE |
|---|---|---|---|---|---|---|
| rsLoRA | 5E-3 | 5E-3 | 1E-2 | 2E-3 | 5E-3 | 2E-3 |
| FFA-rsLoRA | 2E-2 | 2E-2 | 2E-2 | 1E-2 | 2E-2 | 1E-2 |
| FedDPA-reLoRA | 5E-3 | 5E-3 | 1E-2 | 1E-3 | 5E-3 | 1E-2 |
| FedSA-rsLoRA | 5E-3 | 5E-3 | 5E-3 | 1E-3 | 2E-3 | 2E-3 |

Table 8: The learning rates used for VeRA-based methods on the GLUE benchmark.

| Method | Position | MNLI-m | MNLI-mm | SST2 | QNLI | QQP | RTE |
|---|---|---|---|---|---|---|---|
| VeRA | VeRA | 1E-2 | 1E-2 | 2E-2 | 2E-3 | 2E-3 | 1E-2 |
| | Head | 6E-3 | 6E-3 | 2E-3 | 3E-4 | 3E-4 | 2E-4 |
| FFA-VeRA | VeRA | 2E-2 | 2E-2 | 1E-2 | 1E-2 | 1E-2 | 1E-2 |
| | Head | 2E-3 | 2E-3 | 6E-3 | 2E-4 | 6E-3 | 2E-4 |
| FedDPA-VeRA | VeRA | 1E-2 | 1E-2 | 1E-2 | 2E-3 | 2E-2 | 1E-2 |
| | Head | 6E-3 | 6E-3 | 6E-3 | 3E-4 | 2E-3 | 2E-4 |
| FedSA-VeRA | VeRA | 2E-3 | 2E-3 | 1E-2 | 1E-2 | 2E-3 | 1E-2 |
| | Head | 3E-5 | 3E-5 | 3E-4 | 3E-4 | 3E-4 | 1E-4 |

Table 9: Performance of different methods on the GSM8K dataset and example answers generated by each method.

| | LoRA | FFA-LoRA | FedSA-LoRA |
|---|---|---|---|
| Accuracy | 46.24 | 46.32 | **46.63** |
| Question: | In a student council election, candidate A got 20% of the votes while candidate B got 50% more than candidate A's votes. The rest of the votes was given to candidate C. If there were 100 voters, how many votes did candidate C get? | | |
| Output: | Candidate A got 20% of the votes. That is 20% of 100 = 20% × 100 = 20 × 100 = 2000 votes. Candidate B got 50% more than candidate A. That is 50% of 2000 = 50% × 2000 = 50 × 2000 = 10000 votes. The rest of the votes was given to candidate C. So there are 10000 + 2000 = 12000 votes in total. Candidate C got 10000 votes. The answer is 10000. [✗] | Candidate A got 20% of 100 = 100 × 20% = 100 × 20/100 = 20. Candidate B got 20% more than candidate A's votes, which is 20% + 20% = 40%. 40% of 100 = 100 × 40% = 100 × 40/100 = 40. So the two candidates together got 20 + 40 = 60. The rest of the votes were given to candidate C. So candidate C got 100 - 60 = 40. The answer is 40. [✗] | Candidate A got 20% of the votes or 20% * 100 = 20 votes. Then candidate B got 50% more than candidate A's 20 votes so 50% of 20 is 20 * 50% = 10. Then candidate B got 20 + 10 = 30 votes. So candidate C got 100 - 20 - 30 = 50 votes. The answer is 50. [✔] |

### A.6 Learned Matrices Comparison

#### A.6.1 Learned and Initialized LoRA Matrices Comparison

Since the learned LoRA $A$ matrices are similar across different clients in Figure 2, we separately plotted the relationships of $A$ from Figure 2 in Figure 3 to show the relationships in $A$ more clearly. This shows they are similar but not identical. Moreover, we further illustrate the difference between the learned and initialized $A$ matrices for each client under the IID partition in this section. The results, shown in Figure 4, confirm that the $A$ matrices are updated.

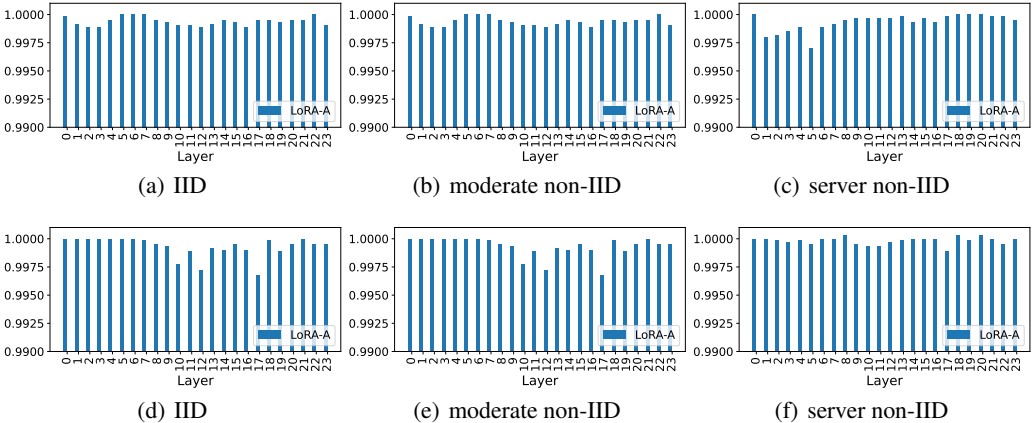

Figure 3: Mean of pairwise cosine similarity of the learned $A$ matrices across layers of a RoBERTa model locally fine-tuned with LoRA on the RTE task, with different levels of data heterogeneity. (a)-(c): value matrices; (d)-(f): query matrices. The learned $A$ matrices across client are similar but not identical.

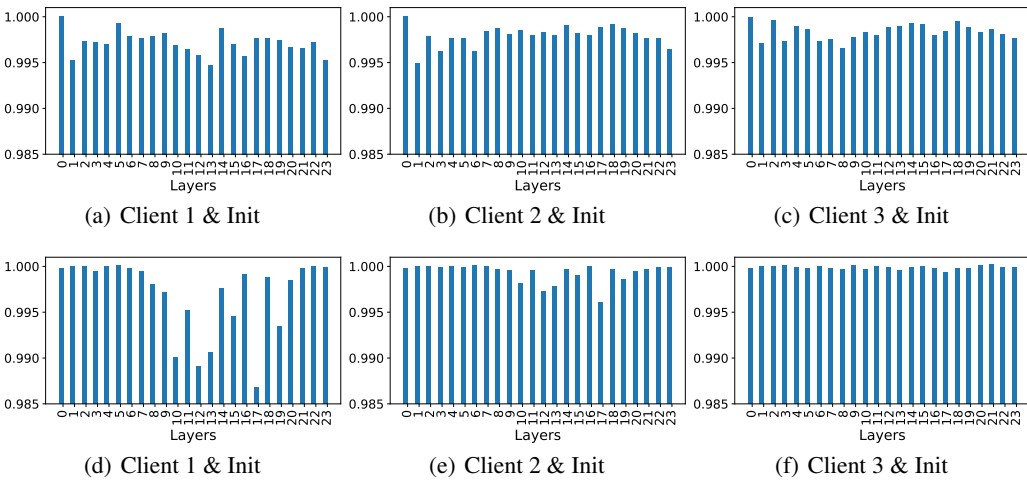

Figure 4: Cosine similarity of learned and initialized $A$ matrices across layers of different clients of a RoBERTa model locally fine-tuned with LoRA on the RTE task. (a)-(c): value matrices; (d)-(f): query matrices. The learned $A$ matrices are different from the initialized $A$ matrices, indicating that the $A$ matrices are updated.

#### A.6.2 Learned rsLoRA Matrices Comparison

In this section, we present the mean of pairwise client relationships for the learned rsLoRA (Kalajdzievski, 2023) matrices. These results, shown in Figure 5, demonstrate a similar phenomenon to the learned LoRA matrices. That is, the learned $A$ matrices are more similar across clients than the

$B$ matrices, and with increased data heterogeneity, the similarity of $B$ matrices between different clients decreases.

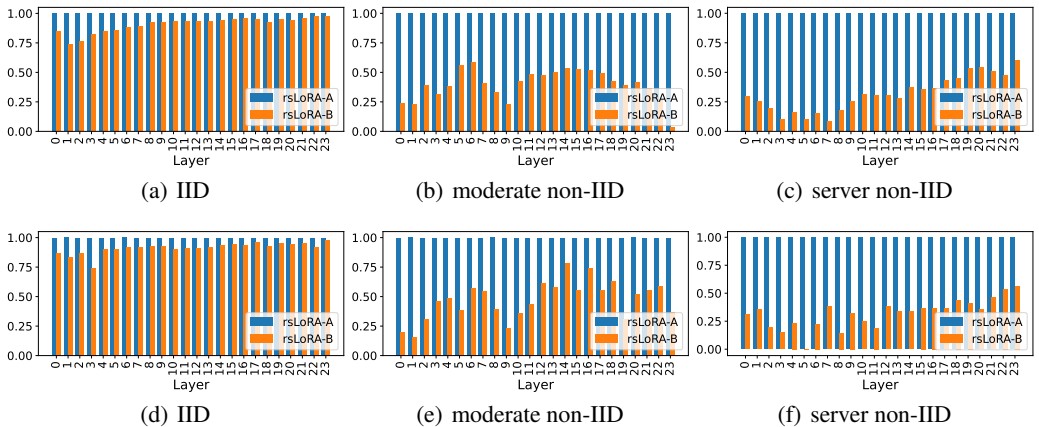

Figure 5: Mean of pairwise cosine similarity of the learned $A$ and $B$ matrices across layers of a RoBERTa model locally fine-tuned with rsLoRA on the RTE task, with different levels of data heterogeneity. (a)-(c): value matrices; (d)-(f): query matrices. The learned $A$ matrices are more similar across clients than the $B$ matrices, and with increased data heterogeneity, the similarity of $B$ matrices between different clients decreases.

### A.6.3 LEARNED VeRA MATRICES COMPARISON

In this section, we show the mean of pairwise client relationships for the learned VeRA (Kopiczko et al., 2024) matrices. In VeRA, the low-rank matrices $A$ and $B$ are initialized using the uniform version of Kaiming initialization, fixed, shared across all layers, and adapted with trainable scaling vectors $d$ and $b$. The $b$ vectors are initialized to zero, and the $d$ vectors are initialized with a value of 0.1. To make the notation consistent with our work, we rewrite the scaling vectors $d$ and $b$ as $A_d$ and $B_b$ to reflect the position of each scaling vector. These results, illustrated in Figure 6, demonstrate a similar phenomenon to the learned LoRA matrices. That is, the learned scaling vectors $A_d$ are more similar across clients than the scaling vectors $B_b$, and with increased data heterogeneity, the similarity of scaling vectors $B_b$ between different clients decreases.

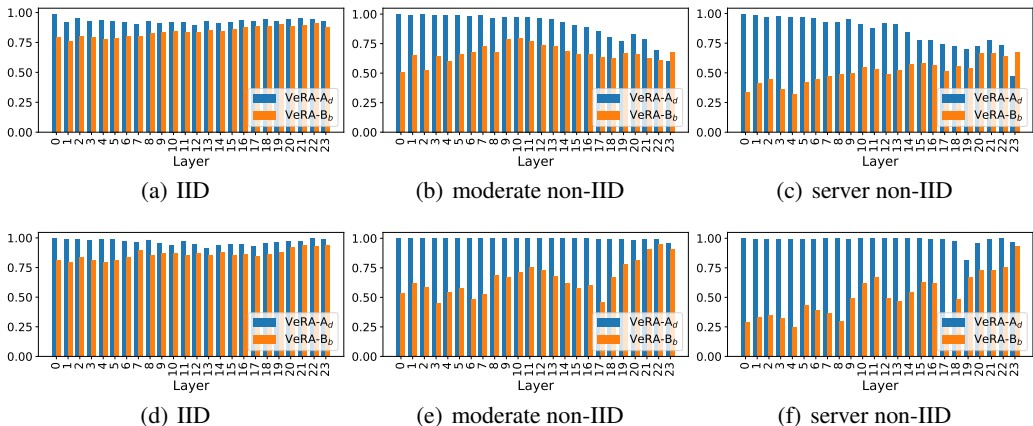

Figure 6: Mean of pairwise cosine similarity of the learned scaling vectors $A_d$ and $B_b$ across layers of a RoBERTa model locally fine-tuned with VeRA on the RTE task, with different levels of data heterogeneity. (a)-(c): value matrices; (d)-(f): query matrices. The learned scaling vectors $A_d$ are more similar across clients than the scaling vectors $B_b$, and with increased data heterogeneity, the similarity of scaling vectors $B_b$ between different clients decreases.

## A.7 FURTHER RESULTS ON COMPRESSING $A$ MATRICES

Based on the reviewers' comments during the review stage, we further conduct experiments on compressing the $A$ matrices. Specifically, we chose FetchSGD (Rothchild et al., 2020) to compress around 50% of the $A$ matrices, given that the updates across the $A$ matrices are trivial but necessary. The experimental results are shown in Table 10, which indicates that FedSA-LoRA with a compression method achieves performance comparable to the original FedSA-LoRA. This confirms that the $A$ matrices can indeed be further compressed to reduce communication overhead while maintaining satisfactory performance, validating the reviewer's hypothesis that "there must exist a compression method or a sparse structure to significantly reduce the number of parameters in the $A$ matrices that need to be updated".

Table 10: Time and space costs for each method on the RTE and QNLI tasks. # Communication round denotes the number of communication rounds to reach the predefined target performance. [‡] denotes equipped with the compressing method FetchSGD.

| | # Trainable Parm. | # Per-round Communicated Parm. | # Per-round Computation Cost | | # Communication Round | | Accuracy | |
|---|---|---|---|---|---|---|---|---|
| | | | RTE | QNLI | RTE | QNLI | RTE | QNLI |
| LoRA | 1.83M | 0.78M | 22s | 35s | 167 | 397 | $87.49_{\pm 0.15}$ | $91.16_{\pm 0.72}$ |
| FFA-LoRA | 1.44M | 0.39M | 20s | 33s | 229 | 374 | $86.08_{\pm 1.16}$ | $91.52_{\pm 0.59}$ |
| FedDPA-LoRA | 2.62M | 0.78M | 23s | 37s | 128 | 325 | $87.44_{\pm 0.13}$ | $90.74_{\pm 1.38}$ |
| FedSA-LoRA | 1.83M | 0.39M | 22s | 34s | 91 | 224 | $87.93_{\pm 0.11}$ | $92.13_{\pm 0.24}$ |
| FedSA-LoRA[‡] | 1.83M | 0.20M | 22s | 34s | 79 | 155 | $87.88_{\pm 0.13}$ | $92.16_{\pm 0.20}$ |

