# OpenReview forum: "Selective Aggregation for Low-Rank Adaptation in Federated Learning"
_ICLR.cc/2025/Conference — ICLR 2025 Poster_

### Official Review · Reviewer_dqpx · 2024-10-18

**Soundness:** 3
**Presentation:** 2
**Contribution:** 2
**Rating:** 6
**Confidence:** 3

**Summary:**

This paper introduces a new way of combining LoRA in the federated setting, called FedSA-LoRA, where the authors found that A matrics are responsible for learning general knowledge, and B matrics focus on client-specific knowledge. Only A matrics are used for server aggregation. Extensive experiment results indicate the effectiveness of the proposed FedSA-LoRA with various LoRA-related variants.

**Strengths:**

- Observing the roles that matrices A and B play is intuitive and important, and this observation is also prevalent in several LoRA-related variants.
- The experiments are comprehensive, covering different non-iid levels and client numbers and across different tasks. Also, the authors provide many ablation studies.
- Authors provide some theoretical justifications for the convergence of their method.

**Weaknesses:**

- Motivation for exploring LoRA in federated learning: Why do authors use and investigate Lora-based methods in FL? The authors can expand and explain more about why it is good compared with parameter-efficient methods. In the introduction and related works, the authors state they are efficient, effective, and flexible, which is a bit too general from my perspective. The authors can better motivate the importance of this problem.
- Line 52-71: First, I feel that what kind of aggregation errors is unclear to me: could the authors provide a more accurate and specific illustration of the aggregation errors? Secondly, I wonder in practice, why do people not do the ideal model update? What are the difficulties in doing that?
- Authors can provide some computational analysis (e.g. time and space cost) of their method compared with the baselines.
- The authors could include standard deviations or confidence intervals for their main results tables, particularly for the main results.
- On the natural language generation task, the improvement of the proposed method seems to be a little marginal. Also, it is tested under IID distribution.
- The overall contribution seems to be marginal from my perspective.

**Questions:**

- The authors mention in the experiment section that the proposed method is more effective when non-IID conditions are more severe. Could the authors provide some potential reasons for that?

---

> ### Author Response · Authors · 2024-11-22
> **Response to Reviewer dqpx (1 / 2)**
>
> We thank the reviewer for the time and effort in reviewing our paper and providing constructive comments. We have modified the motivation for exploring LoRA in federated learning, added explanations about the derived aggregation errors, provided computational analysis, included results from multiple runs, and added more generative tasks in our revision. Please see our response below regarding the specific comments.
>
> > **W1**: motivation for lora-based method
>
> Thank you for the constructive comments. We have modified the writing in the introduction and related works in our revision. In detail, we modified the original sentences
>
> ''Among these, LoRA-based methods have gained significant attention due to their efficiency, effectiveness, and flexibility, which is also the focus of our work.'' to
>
> ''Among these, LoRA-based methods have become increasingly popular, leveraging the assumption that over-parameterized models have a low intrinsic dimension [1, 2]. A pre-trained model can be shared and utilized to create multiple small LoRA modules tailored for different tasks, making them more effective and flexible. Moreover, this simple design allows us to merge the trainable matrices with the frozen weights during deployment, introducing no inference latency. Given these advantages, we focus on LoRA-based methods in this work.''
>
> We hope that the modified version can address the reviewer’s concerns.
>
> > **W2**: what are aggregation errors and why not does the ideal model update?
>
> Thank you for the question. We are glad to provide more explanation about what aggregation errors are and why people do not achieve the ideal model update. When introducing LoRA into FL, the update for the $i$-the client is given by $\Delta_{W_i} =  B_i A_i$. The ''ideal'' model update for server aggregation should be:
> $$
> \Delta_{W}^* = \frac{1}{m}\sum_{i=1}^{m}\Delta_{W_i} = \frac{1}{m}\sum_{i=1}^{m}B_i A_i = \frac{1}{m}(B_1A_1 + B_2A_2 + \cdots + B_mA_m).
> $$
> However, in practical LoRA training, the matrices $A_i$ and $B_i$, not the original $W_i$, are trainable. Thus, we cannot directly average the $\Delta_{W_i}$; instead, we can only separately average $A_i$ and $B_i$, and then combine them to obtain the update:
> $$
> \Delta_{W} =  (\frac{1}{m}\sum_{i=1}^{m}B_i) (\frac{1}{m}\sum_{i=1}^{m}A_i)
> $$
> $$
> = \frac{1}{m}(B_1 + B_2 + \cdots + B_m)\frac{1}{m}(A_1 + A_2 + \cdots + A_m)
> $$
> $$
> =  \frac{1}{m^2} (B_1A_1 + B_1A_2 + \cdots + B_1A_m + B_2A_1 + B_2A_2 + \cdots + B_2A_m + \cdots + B_mA_1 + B_mA_2 + \cdots + B_mA_m),
> $$
> which differs from the ''ideal'' model update. This difference introduces aggregation errors.
>
> Regarding why the ideal model update is not achieved, it is because the matrices $A_i$ and $B_i$ are trainable, not the original $W_i$. While it is possible to first combine $A_i$ and $B_i$ to obtain $\Delta_{W_i}$ and then average them to achieve the "ideal" update, we cannot decompose $W$ back into $A$ and $B$ for further federated training. This limitation prevents people from performing the ideal model update.
>
> We hope this explanation can address the reviewer’s concerns.
>
> > **W4**: The authors could include standard deviations or confidence intervals for their main results tables, particularly for the main results.
>
> Thank you for the constructive comments. As recommended by the reviewer, we further conducted multiple runs to compute the means and standard deviations for each method. We refer to Table A1  in our **Response to All** for these new results. These results demonstrate the stability and effectiveness of the proposed method. We hope the added multiple experiments can address the reviewer’s concerns.

---

> ### Author Response · Authors · 2024-11-22
> **Response to Reviewer dqpx (2 / 2)**
>
> > **W3**: computational analysis (e.g. time and space cost)
>
> Thank you for the constructive comments. We have provided additional system efficiency analysis (including computational analysis) in the revision. The system efficiency in FL consists of communication cost and computation cost. To provide a comprehensive comparison, we detail the number of trainable parameters, the number of communication model parameters per FL round, the computation cost per FL round, and the number of communication rounds needed to reach the predefined target performance on the RTE and QNLI tasks in Table A4. The target performance is defined as 95\% of the prediction accuracy provided in LoRA [3]. Specifically, we define the target performance of the RTE and QNLI tasks as 80.94\% and 90.06\%, respectively.
>
> As shown in below table, our FedSA-LoRA requires the smallest communication cost to reach the target performance. Communication cost is a critical factor in FL, as it significantly impacts the overall system efficiency. The communication cost can be roughly estimated by considering the number of transmitted messages required to achieve a target performance, calculated as *\# transmitted messages= \# communication round $\times$ \# communicated model parameter*. Additionally, while our FedSA-LoRA requires more trainable model parameters and incurs slightly more computation cost per FL round than the baseline FFA-LoRA (22s compared to 20s on RTE task), it is important to note that our model reaches the target performance with fewer communication rounds (\#91 compared to \#229 on RTE task).  These computation and communication costs demonstrate the overall efficiency of our model.
>
> We hope that the added system efficiency analysis can address the reviewers’ concerns.
>
> || Trainable Parm.|# Per-round Communicated Parm. |# Per-round Computation Cost (RTE)|# Per-round Computation Cost (RNLI)|# Communication Round (RTE)|# Communication Round (QNLI) |
> |-|-|-|-|-|-|-|
> |LoRA|1.83M|0.78M|22s|35s|167|397|
> |FFA-LoRA|1.44M|0.39M|20s|33s|229|374|
> |FedDPA-LoRA|2.62M|0.78M|23s|37s|128 |325|
> |FedSA-LoRA|1.83M|0.39M|22s|34s|91|224|
> #### Table A4. Time and space costs for each method on the RTE and QNLI tasks. \# Communication round denotes the number of communication rounds to reach the predefined target performance.
>
>
> > **W5**: On the natural language generation task, the improvement of the proposed method seems to be a little marginal. Also, it is tested under IID distribution.
>
> Yes, due to the characteristics of this dataset, we could only perform IID partitioning, which resulted in our method showing limited improvement on this dataset. As shown in the Effect of Data Heterogeneity under the In-depth Analyses in our paper, our method performs better in non-IID scenarios. Therefore, we further conducted experiments on a code generation task that allows for non-IID partitioning. We chose the CodeSearchNet dataset [4] and used the default non-IID partitioning provided in FederatedScope-LLM [5]. The performance scores for LoRA, FFA-LoRA, and our FedSA-LoRA are 58.34, 58.57, and 59.66, respectively, which further validates the effectiveness of our method in generation tasks. We have included these results in the revision and hope that the added results can address the reviewer’s concerns.
>
> > **Q1**: why better on non-IID?
>
> Thank you for the valuable question. This phenomenon is consistent with our Figure 2, which shows that with increased data heterogeneity, the similarity of $B$ matrices between different clients decreases. Therefore, when the non-IID conditions are more severe, the advantages of keeping $B$ locally become more pronounced. In this case, the learned $B$ will be less similar, highlighting the need for personalization. We have added this explanation in our revision and hope that it can address the reviewer’s concerns.
>
> [1] Li, Chunyuan, et al. "Measuring the intrinsic dimension of objective landscapes." arXiv preprint arXiv:1804.08838 (2018).
>
> [2] Aghajanyan, et al. "Intrinsic Dimensionality Explains the Effectiveness of Language Model Fine-Tuning." ACL-IJCNLP. 2021.
>
> [3] Hu, Edward J., et al. "Lora: Low-rank adaptation of large language models." arXiv preprint arXiv:2106.09685 (2021).
>
> [4] Husain, Hamel, et al. "Codesearchnet challenge: Evaluating the state of semantic code search." arXiv preprint arXiv:1909.09436 (2019).
>
> [5] Kuang, Weirui, et al. "Federatedscope-llm: A comprehensive package for fine-tuning large language models in federated learning." KDD. 2024.

---

> > ### Comment · Reviewer_dqpx · 2024-11-25
> >
> > Thanks for the additional experiments, analysis, and explanations addressing my concerns. I am willing to increase my score.

---

> > > ### Author Response · Authors · 2024-11-26
> > > **Thanks for Reply**
> > >
> > > Many thanks to the reviewer for their thoughtful consideration and willingness to increase the score. We are pleased that our response has addressed the reviewer’s concerns and sincerely appreciate their recognition of our efforts, as well as the constructive comments that have helped us improve our work.

---

### Official Review · Reviewer_Lpwz · 2024-10-30

**Soundness:** 2
**Presentation:** 3
**Contribution:** 2
**Rating:** 5
**Confidence:** 3

**Summary:**

The paper begins by analyzing the roles of matrices A and B in LoRA, concluding that matrix A is responsible for general knowledge while matrix B handles domain-specific knowledge. Based on this analysis, the authors propose FedSA-LoRA, a method that aggregates only the A matrix during federated learning while retaining the edge-side B matrix. This approach aims to integrate general capabilities while preserving personalized knowledge.

**Strengths:**

(1) The paper's analysis of the distinct roles of matrices A and B clearly demonstrates its insights and motivations.

(2) The proposed method improves upon existing solutions and achieves certain performance enhancements.

**Weaknesses:**

(1) One of the significant contributions of the paper is the asymmetric analysis of matrices A and B, concluding that matrix A is responsible for general knowledge and matrix B for domain-specific knowledge. However, this perspective has already been proposed in several works, such as HydraLoRA [1], which also designed an asymmetric LoRA structure based on this concept. The paper should properly cite relevant literature, and these existing findings somewhat diminish the paper's contribution.

(2) The experimental results are based on a single run, lacking multiple experiments to calculate averages and perform variance analysis, which may affect the reliability of the results. It is recommended that the authors refer to previous related work [2], conduct multiple experiments, and compute means and variances to more comprehensively evaluate the method's stability and effectiveness.

(3) The paper mainly considers the sharing of edge-side models but lacks a reasonable mechanism to update the server-side model to endow it with comprehensive capabilities.

[1] HydraLoRA: An Asymmetric LoRA Architecture for Efficient Fine-Tuning
[2] Improving LoRA in Privacy-preserving Federated Learning

**Questions:**

(1) In the experimental results, why does FedSA-LoRA outperform other methods under non-heterogeneous conditions? In the absence of heterogeneity, since both matrices A and B can learn shareable knowledge, sharing information from both matrices—or sharing matrix B's information as in FFA-LoRA—should intuitively achieve equal or even greater gains.

---

> ### Author Response · Authors · 2024-11-22
> **Response to Reviewer Lpwz (1 / 2)**
>
> We thank the reviewer for the time and effort in reviewing our paper and providing constructive comments. In our revision, we have added the suggested reference and results of multiple runs, elaborated on the differences between our work and related studies, and addressed the concern regarding the absence of a server-side model and experiments in non-IID settings. Please see our response below regarding the specific comments.
>
> > **W1**: related work [1]
>
> We thank the reviewer for pointing out the work [1] and for the valuable suggestion. We agree that the asymmetry analyses of $A$ and $B$ in LoRA is not our original contribution, as these have been explored in previous works such as the one we cited [3] and the reviewer's referenced work [1]. We acknowledge that we previously overlooked this work.  In our revised manuscript, we have cited [1] and added a detailed discussion to clarify the differences between our work and the existing similar work for the asymmetry analyses of $A$ and $B$ in LoRA. Below is a summary of the unique contributions of our paper:
>
> 1) Our paper distinctly considers the asymmetry analyses of $A$ and $B$ within LoRA in the context of federated related, while these related papers [1, 3] can further serve as verification of the effectiveness of our method. In particular, we further analyzed different non-IID scenarios, as data heterogeneity is a significant issue in federated learning. Through our analysis, we not only found that the learned $A$ matrices are more similar across clients than the $B$ matrices (consistent with previous findings), but more importantly, we discovered that with increased data heterogeneity, the similarity of $B$ matrices between different clients decreases. This is a specific new finding in the context of federated learning.
>
> 2) We extended this asymmetry analysis to other LoRA variants, such as rsLoRA and VeRA, and found similar phenomena. This generalization was previously lacking in the literature [1, 3].
>
> We would like to thank the reviewer once again for pointing out this reference [1] and for the valuable suggestions. We have incorporated the above discussions into our revision for further clarification. We hope this explanation addresses the reviewer’s concerns.
>
>
> > **W2**: multiple runs
>
> Thank you for the constructive comments. As recommended by the reviewer, we further conducted multiple experiments to compute the means and standard deviations to more comprehensively evaluate the method's stability and effectiveness. We refer to Table A1 in our **Response to All** for these new results. These results demonstrate the stability and effectiveness of the proposed method. We hope the added multiple experiments can address the reviewer’s concerns.
>
> > **W3**: The paper mainly considers the sharing of edge-side models but lacks a reasonable mechanism to update the server-side model to endow it with comprehensive capabilities.
>
> As shown in lines 81-85 in our paper, our work belongs to the personalized FL category, an active area of research in FL that supports personalization solutions and tackles data heterogeneity challenges. The core objective of personalized FL is to produce diverse models that can adapt to the specific needs of each client, considering variations in data distributions, user preferences, and usage patterns. Our experiments follow existing personalized FL work [4,5,6], aiming to achieve models with better personalization performance.
> We have added modifications to further clarify that our work belongs to personalized FL in revision. Please see the added description in the introduction (lines 111-112 in our revision).
>
> How to obtain a strong global model is beyond the scope of this work. However, we recognize its importance and are interested in exploring this in future research. For example, we could utilize an additional global validation set that covers the data distribution of all clients to compare the performance of each personalized model and choose the best one as the final global model.
> Thank you for the reviewer’s comments. We hope this explanation can address the reviewer's concerns.

---

> ### Author Response · Authors · 2024-11-22
> **Response to Reviewer Lpwz (2 / 2)**
>
> > **Q1**: why does FedSA-LoRA outperform other methods under non-heterogeneous conditions?
>
> As shown in Figure 2 in our paper, even under non-heterogeneous conditions, i.e., IID, the learned $B$ matrices are still less similar across clients than the $A$ matrices. This is because, even in IID conditions, the specific data for each client is different, even though they follow the same distribution. Since $B^*$ is related to the input data, as illustrated in Lemma 1, the learned $B$ matrices will still exhibit some dissimilarity across clients. Thus, keeping a personalized $B$ is still preferable in this scenario, although this advantage might not be as evident as in non-IID cases.
>
> Moreover, since vanilla LoRA, which aggregates both $A$ and $B$, introduces aggregation errors, and FFA-LoRA, which freezes the $A$ matrices, impairs the learning ability of LoRA, both approaches can result in suboptimal performance. In contrast, our FedSA-LoRA shares only the $A$ matrices with the server for aggregation while keeping the $B$ matrices local, achieving superior performance compared to these methods.
>
> We hope this explanation can address the reviewer's concerns.
>
> [1] HydraLoRA: An Asymmetric LoRA Architecture for Efficient Fine-Tuning
>
> [2] Improving LoRA in Privacy-preserving Federated Learning
>
> [3] Asymmetry in low-rank adapters of foundation models
>
> [4] Personalized federated learning with moreau envelopes
>
> [5] Exploiting Shared Representations for Personalized Federated Learning
>
> [6] Personalized Federated Learning with Feature Alignment and Classifier Collaboration

---

> > ### Comment · Reviewer_Lpwz · 2024-11-25
> >
> > Thank you for the response; it has partially addressed my concerns. Based on my overall assessment of the paper, I prefer to maintain my current rating.

---

> > > ### Author Response · Authors · 2024-11-25
> > > **Thanks for Reply**
> > >
> > > Many thanks for the reviewer's reply. We would appreciate it if the reviewer could let us know what specific concerns remain unresolved, and we would be glad to continue discussing them with the reviewer.

---

> > > ### Author Response · Authors · 2024-12-02
> > > **Looking Forward to Reviewer Lpwz’s Feedback**
> > >
> > > Dear Reviewer Lpwz,
> > >
> > > We appreciate the constructive comments in your initial reviews, and we sincerely thank you for the time and effort you have dedicated to reviewing our paper. As the deadline for discussion approaches, we would be grateful if you could help identify any specific concerns that remain unresolved in our manuscript. We would be glad to continue discussing them.
> > >
> > > Best regards, \
> > > Authors

---

### Official Review · Reviewer_tZCA · 2024-11-02

**Soundness:** 3
**Presentation:** 3
**Contribution:** 2
**Rating:** 8
**Confidence:** 4

**Summary:**

In this paper, the authors present FedSA-LoRA, a novel approach for federated learning that shares only the A matrices across clients to leverage the functional differences between A and B matrices in LoRA modules. The method’s effectiveness is validated through both theoretical analysis and empirical results.

**Strengths:**

1. The paper is generally well-motivated and easy to follow.
2. This work introduces the shared-A LoRA technique into the FL framework, making it promising compared to previous work where the down-projection matrix (A matrix) is kept frozen.
3. Comprehensive experimental results are provided.

**Weaknesses:**

1. While this work introduces a shared-A LoRA framework into FL, such a technique has already been introduced in the MoE area [1] under similar motivations. Although the novelty of introducing it into FL should be recognized, the introduction of such a framework cannot be solely credited to this work.

2. On page 20, Figure 3, the authors claim that "A matrices are different from the initialized A matrices, indicating that the A matrices are updated." However, the cosine similarity between the learned and initialized down-projection matrices remains quite high (greater than 0.985), which indicates that the A matrices have changed very little compared to the initialization.

3. Compared with [2], the cosine similarity across A matrices for different clients reported in this work is very high. Why does this happen? Is it caused by different levels of data heterogeneity?

4. Assumption 3, especially the last two inequalities, lacks verification and explanation. It appears these are purely for simplifying the proof of Theorem 1 on page 16. The authors need to elaborate more on this uncommon assumption, including under what conditions it could hold and what properties of the A and B matrices this assumption reveals.

5. In the provided code, alongside the B matrices being personalized, a classifier layer is also personalized. Is this necessary? How does this personalized classifier layer influence the experimental results?

References:

[1] Tian, Chunlin, et al. "HydraLoRA: An Asymmetric LoRA Architecture for Efficient Fine-Tuning." arXiv preprint arXiv:2404.19245 (2024).

[2] Zhu, Jiacheng, et al. "Asymmetry in low-rank adapters of foundation models." arXiv preprint arXiv:2402.16842 (2024).

**Questions:**

See above. I will increase my score if my concerns are well addressed.

---

> ### Author Response · Authors · 2024-11-22
> **Response to Reviewer tZCA (1 / 2)**
>
> We thank the reviewer for the time and effort in reviewing our paper and providing constructive comments. We are pleased that the reviewer found the paper to be ''well-motivated'', ''easy to follow'', ''promising'', and ''comprehensive'', which inspires us a lot. We have added the suggested reference, detailed our differences with these related works, and more explanations about our assumptions in our revision. Please see our response below regarding the specific comments.
>
> > **W1**: related work in the MoE area [1]
>
> We thank the reviewer for pointing out the work in the MoE area [1] and for the valuable suggestion. We agree that the asymmetry analysis of $A$ and $B$ in LoRA is not our original contribution, as these have been explored in previous works such as the one we cited [2] and the reviewer's referenced work [1]. We acknowledge that we previously overlooked this work. In our revised manuscript, we have cited [1] and added a detailed discussion to clarify the differences between our work and the existing similar work for the asymmetry analyses of $A$ and $B$ in LoRA. Below is a summary of the unique contributions of our paper:
>
> 1) Our paper distinctly considers the asymmetry analyses of $A$ and $B$ within LoRA in the context of federated related, while these related papers [1, 2] can further serve as verification of the effectiveness of our method. In particular, we further analyzed different non-IID scenarios, as data heterogeneity is a significant issue in federated learning. Through our analysis, we not only found that the learned $A$ matrices are more similar across clients than the $B$ matrices (consistent with previous findings), but more importantly, we discovered that with increased data heterogeneity, the similarity of $B$ matrices between different clients decreases. This is a specific new finding in the context of federated learning.
>
> 2) We extended this asymmetry analysis to other LoRA variants, such as rsLoRA and VeRA, and found similar phenomena. This generalization was previously lacking in the literature [1, 2].
>
> We would like to thank the reviewer once again for pointing out this reference [1] and for the valuable suggestions. We have incorporated the above discussions in our revision for further clarification. We hope this explanation addresses the reviewer’s concerns.
>
> > **W2**: On page 20, Figure 3, A matrices have changed very little compared to the initialization.
>
> Yes, as pointed out by the reviewer and shown in Figure 3 in our paper (now it's Figure 4 in our revision), the $A$ matrices have changed little compared to the initialization. This is also why previous works proposed fixing the $A$ matrices in LoRA to further reduce memory cost [2, 3]. However, fixing the $A$ matrices in LoRA can impair its learning ability, leading to suboptimal performance [3]. This is also validated in our experiments by comparing the performance of FFA-LoRA with our FedSA-LoRA. Therefore, we believe that the $A$ matrices still need to be learnable, even though their changes are minimal. We hope this explanation can address the reviewer’s concerns.
>
> > **W3**: different similarity of A matrices to [2]
>
> We apologize for any confusion and would like to clarify that the similarity metric used in [2] and in our work is different. Specifically, the similarity metric used in [2] is the canonical correlation analysis goodness of fit [4], whereas we use cosine similarity in our work. Due to the high computational cost of the canonical correlation analysis goodness of fit employed in [2], we adopted the widely used cosine similarity as the similarity metric. We hope this explanation can address the reviewer’s concerns.

---

> ### Author Response · Authors · 2024-11-22
> **Response to Reviewer tZCA (2 / 2)**
>
> > **W4**: more explanations about Assumption 3.
>
> Thank you for the valuable suggestions. We have provided more explanation about our assumptions in our revision, and below are the details.
>
> According to the definition of the Frobenius norm $ \||A\||_{\text{F}}=\sqrt{\sum_i^m \sum_j^n|a\_{i j}|^2} $, we know that the first two inequalities in Assumption 3 hold when all parameter values in $A_i^{(t)}$ and $B_i^{(t)}$ are finite. If we denote the eigenvalues of $A\_i^{(t)}$ by $\{\lambda\_{1}, \ldots, \lambda\_{K}\}$, then the eigenvalue of $(A\_i^{(t)})^{\top} A\_i^{(t)}$ are $\{\lambda^2\_{1},\ldots,\lambda^2\_{K}\}$. Similarly, if the eigenvalues of $\nabla \mathcal{L}\_i(W\_i^{(t)})$ are $\{\mu\_{1},\ldots,\mu\_{K}\}$, then the eigenvalues of $(\nabla \mathcal{L}\_i(W\_i^{(t)}))^{\top}\nabla \mathcal{L}\_i(W\_i^{(t)})$ are $\{\mu^2\_{1},\ldots,\mu^2\_{K}\}$. Noting that $\langle {A\_i^{(t)}}^{\top}A\_i^{(t)}, \nabla \mathcal{L}\_i(W\_i^{(t)})^{\top}\nabla \mathcal{L}\_i(W\_i^{(t)})\rangle\_F = \sum\_{k}\lambda\_k^2 \mu\_k^2$, and $\||\nabla \mathcal{L}\_i(W\_i^{(t)})\||\_F^2 = \sum\_{k}\mu\_k^2$, the third inequality of Assumption 3 holds when there exist a constant $c_A$ such that $\lambda_k^2 \geq c_A, \forall k$. In other words, the third inequality of Assumption 3 holds when all the eigenvalues of $A\_i^{(t)}$ are non-zero (choose $c_A = \min\{|\lambda_1|,\ldots,|\lambda_K|\}$). Similarly, the last inequality of Assumption 3 holds when all the eigenvalues of $B_i^{(t)}$ are non-zero.
>
> We have added this explanation in our revision and hope that it can address the reviewer’s concerns.
>
> > **W5**: why personalized classifier layer?
>
> Thank you for the reviewer's question. The personalized classifier layer designed in our work is motivated by previous works [5, 6], which suggest that a personalized classifier is better for federated learning, especially in non-IID scenarios. We also conducted simple experiments on our datasets that verify this, which is why we adopted the personalized classifier in our work. Note that for a fair comparison, all the baselines also adopted the personalized classifier in this work.
>
> To comprehensively show this, we further conducted multiple runs to compute the mean and standard deviation; the results are shown in below table. It indicates that the personalized classifier is indeed better, but not by a large margin. Furthermore, since the personalized classifier can further reduce communication costs, we chose it for this work. We hope this explanation can address the reviewer’s concerns.
>
> |        | LoRA $^\dagger$   | LoRA $^{\dagger\dagger}$ | FFA-LoRA $^{\dagger}$ | FFA-LoRA $^{\dagger\dagger}$ | FedSA-LoRA $^{\dagger}$ | FedSA-LoRA $^{\dagger\dagger}$ |
> |--------|---------------------|--------------------------|-----------------------|------------------------------|-------------------------|-------------------------------|
> | SST2   | $95.03_{\pm 0.08}$  | $95.16_{\pm 0.09}$       | $94.90_{\pm 0.04}$    | $94.95_{\pm 0.04}$           | $95.89_{\pm 0.05}$      | $96.00_{\pm 0.04}$            |
> #### Table A3. Comparison of with and without the personalization classifier layer. $^{\dagger}$ denotes without the personalization classifier layer. $^{\dagger\dagger}$ denotes with the personalization classifier layer, which is chosen in this work.
>
> [1] Tian, Chunlin, et al. "HydraLoRA: An Asymmetric LoRA Architecture for Efficient Fine-Tuning." arXiv preprint arXiv:2404.19245 (2024).
>
> [2] Zhu, Jiacheng, et al. "Asymmetry in low-rank adapters of foundation models." arXiv preprint arXiv:2402.16842 (2024).
>
> [3] Zhang, Longteng, et al. "Lora-fa: Memory-efficient low-rank adaptation for large language models fine-tuning." arXiv preprint arXiv:2308.03303 (2023).
>
> [4] Ramsay, James O., et al. "Matrix correlation." Psychometrika 49.3 (1984): 403-423.
>
> [5] Collins, Liam, et al. "Exploiting shared representations for personalized federated learning." International conference on machine learning. PMLR, 2021.
>
> [6] Xu, Jian, et al. "Personalized Federated Learning with Feature Alignment and Classifier Collaboration." The Eleventh International Conference on Learning Representations, 2023.

---

> > ### Comment · Reviewer_tZCA · 2024-11-26
> > **Thank you for the reply**
> >
> > I have decided to increase my score since my major concerns have been addressed in the authors' rebuttal. However, from my understanding, the potential contributions have not been fully investigated. As reported in the main paper and the rebuttal, updating across the A matrices is trivial but necessary. Therefore, there must exist a compression method or a sparse structure to significantly reduce the number of parameters in the A matrices that need to be updated. This aligns with the problem setting in this work since communication overhead is a critical issue in federated learning.
> >
> > Furthermore, if the role of the downstream task is to find the optimal representation (range space), then the updates on the A matrices should be significant. This also raises the question of what the role of A is in this context.
> >
> > I look forward to seeing the authors investigate these directions either in the final revision of this work or in subsequent research. Good luck.

---

> > > ### Author Response · Authors · 2024-11-26
> > > **Thanks for Reply**
> > >
> > > Many thanks to the reviewer for their thoughtful consideration and willingness to increase the score. We are pleased that our response has addressed the reviewer’s major concerns and sincerely appreciate their recognition of our efforts, as well as the constructive comments that have helped us improve our work.
> > >
> > > We would like to thank the reviewer again for the constructive advice on deepening our analysis of the $A$ matrices, which is very meaningful and has sparked our future work. Specifically, as suggested by the reviewer, we will explore how to compress or sparsify the $A$ matrices to decrease communication overhead while maintaining performance, given that the updates across the $A$ matrices are trivial but necessary. This is important since communication overhead is a critical issue in federated learning, and we are eager to investigate it further. Moreover, examining the updates of the $A$ matrices when the downstream task is to find the optimal representation is also an interesting direction, which we will leave as future work. Thanks again for the reviewer's constructive comments, and we hope to discover something new in these directions.

---

> ### Author Response · Authors · 2024-12-02
> **Further Results on Compressing A matrices**
>
> Thanks to the extension of the discussion deadline, we can now present our results on further compressing the $A$ matrices, as suggested by the reviewer's insightful comments. Specifically, we chose FetchSGD [1] to compress around 50\% of the $A$ matrices, given that the updates across the $A$ matrices are trivial but necessary. The table below shows that FedSA-LoRA with a compression method achieves comparable performance to the original FedSA-LoRA. This confirms that the $A$ matrices can indeed be further compressed to reduce communication overhead while maintaining satisfactory performance, validating the reviewer’s hypothesis that "there must exist a compression method or a sparse structure to significantly reduce the number of parameters in the $A$ matrices that need to be updated."
>
> We will incorporate these new results and valuable discussion in our revisions. We greatly appreciate the reviewer’s recognition of our approach based on the asymmetry analysis of the learned $A$ and $B$ matrices in the context of FL, as well as their valuable and constructive advice, which has enhanced our work and sparked ideas for future research.
>
> || \# Trainable Parm.|# Per-round Communicated Parm. |# Per-round Computation Cost (RTE)|# Per-round Computation Cost (RNLI)|# Communication Round (RTE)|# Communication Round (QNLI) | Accuracy (RTE) | Accuracy (QNLI)
> |-|-|-|-|-|-|-|-|-|
> |LoRA|1.83M|0.78M|22s|35s|167|397|$87.49_{\pm 0.15}$ | $91.16_{\pm 0.72}$
> |FFA-LoRA|1.44M|0.39M|20s|33s|229|374|$86.08_{\pm 1.16}$ | $91.52_{\pm 0.59}$
> |FedDPA-LoRA|2.62M|0.78M|23s|37s|128 |325|$87.44_{\pm 0.13}$ | $90.74_{\pm 1.38}$
> |FedSA-LoRA|1.83M|0.39M|22s|34s|91|224|$87.93_{\pm 0.11}$ | $92.13_{\pm 0.24}$
> |FedSA-LoRA $^{\ddagger}$| 1.83M | 0.20M | 22s	| 34s | 79 | 155  | $87.88_{\pm 0.13}$ | $92.16_{\pm 0.20}$
>
> #### Table A5. Time and space costs for each method on the RTE and QNLI tasks. \# Communication round denotes the number of communication rounds to reach the predefined target performance. $^{\ddagger}$ denotes equipped with the compressing method FetchSGD.
>
> ----
> ----
> We forgot to include the reference previously and have added it now.
>
> [1] Rothchild, Daniel, et al. "Fetchsgd: Communication-efficient federated learning with sketching." International Conference on Machine Learning. PMLR, 2020.

---

> > ### Comment · Reviewer_tZCA · 2024-12-02
> > **Thanks for Reply**
> >
> > Thank the authors for the subsequent experiments and discussions, I appreciate it.

---

### Official Review · Reviewer_KuVq · 2024-11-03

**Soundness:** 2
**Presentation:** 3
**Contribution:** 2
**Rating:** 5
**Confidence:** 4

**Summary:**

This paper proposes a low-rank federated fine-tuning algorithm, namely FedSA-LoRA, which regards $A$ as a global adapter and $B$ as a personalized adapter. This algorithm reduces the communication cost by only aggregating matrices $A$.

**Strengths:**

This paper has the interesting motivation and shows the different roles of low-rank matrices $A$ and $B$ in federated fine-tuning.

**Weaknesses:**

I have the following concerns:

1. Lemma 1 is very interesting to me and the verified experiments show that all the matrices $A_i$ ($i$ is the client index) seems to be the same and $B_i$ differs with each other. We know that different client has independent initialization of $A$, but finally with the exactly same $A$ since the similarity of $A$ from clients is $1.0$ in Figure 2. As you mentioned in Figure 3 in Appendix that the learned A matrices are different from the initialized A matrices, so how do you guarantee that the learned A across clients are the totally same even in the server non-IID scenario.

2. The implementation details for Figure 2 are not clear to me, especially how do you implement federated lora fine-tuning, do you update/aggregate both $A$ and $B$?

3. In Assumption 1, what is the meaning of $\mathcal{L}_i(W_j)$ and $\nabla \mathcal{L}_i(W_j)$? In your descriptions, both $i$ and $j$ are the client's index, right? Then how can you access the client-$j$'s gradient information in client $i$?

4. Assumption 2 is too strong, it becomes very easy to handle dissimilarity between clients and stochastic gradients with this assumption. This assumption doesn't appear in smoothness cases in many federated learning papers, so how do you verify this assumption? Can you provide examples to verify the rationality.

5. As I mentioned in concern 4, assumption 2 is a strong assumption, which does not appear in work [1]. Even the same convergence rate as [1] is achieved, that comparison is meaningless to me.

[1] Jianyu Wang, Qinghua Liu, Hao Liang, Gauri Joshi, and H Vincent Poor. Tackling the objective inconsistency problem in heterogeneous federated optimization. Advances in neural information processing systems, 33:7611–7623, 2020.

6. In experiments, they compare with FFA-LoRA, and show better performance than FFA-LoRA. My concern is that, is this a fair comparison? Because the trainable parameters are not the same, FFA-LoRA does not tune the matrices $A$, but FedSA-LoRA does, which means FedSA-LoRA has much more trainable parameters than FFA-LoRA. I hope to see the report of trainable parameters for each algorithm.

7. Many advanced baselines are not compared, so the performance of the proposed algorithm is not convincing. For example, some related work mentioned in the paper, FedLoRA [2], FedDPA [3], and HetLoRA [4].

[2] Liping Yi, Han Yu, Gang Wang, and Xiaoguang Liu. Fedlora: Model-heterogeneous personalized federated learning with lora tuning. arXiv preprint arXiv:2310.13283, 2023. \
[3] Yiyuan Yang, Guodong Long, Tao Shen, Jing Jiang, and Michael Blumenstein. Dual-personalizing adapter for federated foundation models. arXiv preprint arXiv:2403.19211, 2024.\
[4] Yae Jee Cho, Luyang Liu, Zheng Xu, Aldi Fahrezi, Matt Barnes, and Gauri Joshi. Heterogeneous lora for federated fine-tuning of on-device foundation models. In International Workshop on Federated Learning in the Age of Foundation Models in Conjunction with NeurIPS 2023, 2023.

**Questions:**

Please refer to Weakness.

---

> ### Author Response · Authors · 2024-11-22
> **Response to Reviewer KuVq (1 / 2)**
>
> We thank the reviewer for the time and effort in reviewing our paper and providing constructive comments. We have added the suggested baselines, compared parameters, clarified the relationship of the $A$ matrices in Figure 2, and modified the notation in Assumption 1, as well as the convergence rate comparison in our revision. Please see our response below regarding the specific comments.
>
> > **W1 & W2**: confusion about Figure 2.
>
> We apologize for any confusion and would like to provide more explanation about how Figures 2 and 3 (now it's Figure 4 in our revision) are derived and their meanings.
>
> First, we would like to clarify that the matrix $A$ initialized for each client is identical because it is initialized by the server and then sent to each client for training. Regarding the implementation details for Figure 2, we utilize "locally fine-tune" as mentioned in Line 237 in our paper. In this locally fine-tuning setting, both $A$ and $B$ are trained locally by each client and not shared with the server for aggregation. This approach allows us to reveal how the $A$ and $B$ matrices change for different local clients.
>
> Second, the fact that different clients learn similar $A$ matrices is not controlled by us; it's a result of the model's inherent learning process, which aligns with our Lemma 1. In contrast, while the $B$ matrices are also initialized identically for different clients, they eventually diverge and learn distinct $B$ matrices, which further supports our findings in Lemma 1.
>
> Lastly, since Figure 2 shows that the $A$ matrices appear very similar, this may lead readers to argue that the $A$ matrices are not updated at all. We thus provide Figure 3 (now it’s Figure 4 in our revision) to further demonstrate that the $A$ matrices are indeed updated. In short, the $A$ matrices in different clients are indeed learned and updated, but different clients’ $A$ matrices are similar to each other even after training. Furthermore, to show the relationships in $A$ more clearly, we separately plotted the relationships of $A$ from Figure 2 in Figure 3 in Appendix (see our updated revision). This shows they are similar but not identical.
>
> We hope this explanation addresses the reviewer's concerns and helps the reviewer better understand Figures 2 and 3 (now it's Figure 4 in our revision).
>
> > **W3**: notation in Assumption 1.
>
> Thank you for your careful reading and valuable question. We apologize for any confusion. There is a typo in our paper: $W_k$ and $W_j$ should be $W_{i,k}$ and $W_{i,j}$, which represent any different model weights of client $i$. Assumption 1 is a classical definition of $L$-smoothness, widely used in previous works [1, 5, 6, 7]. We hope this explanation can address the reviewer’s concerns, and we have made the necessary modifications in our revision.
>
> > **W4 & W5**: concerns about Assumption 2 and convergence rate compared to [1].
>
> We apologize for any confusion regarding Assumption 2 and our convergence rate compared to work [1]. Work [1] indeed does not use Assumption 2 used in our work, but instead uses an additional bounded dissimilarity assumption on the averaged norm square of local gradients to derive the convergence rate. We understand that it may not be fair to compare our convergence rate with that of work [1], and we have modified the convergence rate comparison in our revision.
>
> Moreover, similar to the  $L$-smoothness Assumption 1, Assumption 2 in our paper (bounded second moment assumption) is also widely used in many federated learning papers [5, 6, 7, 8]. Under this same assumption, we can derive a similar convergence rate to FedAvg. Thank you again for the reviewer pointing this out. We have provided more explanation about the assumptions used in our work below these assumptions and modified the convergence rate comparison in our revision.

---

> ### Author Response · Authors · 2024-11-22
> **Response to Reviewer KuVq (2 / 2)**
>
> > **W6**: concerns regarding whether it's a fair comparison to FFA-LoRA and shows the trainable parameters for each algorithm.
>
> Thank you for the constructive comments. We further compare the trainable parameters and communication parameters for each algorithm in our revision, which are also shown below. It indicates that although our FedSA-LoRA has more trainable parameters than FFA-LoRA, the communication parameters for both are the same. This is important because communication parameters are also a key issue in federated learning.
>
> Regarding the reviewer's concern that the superiority of FedSA-LoRA might be derived from having more trainable parameters, the in-depth analysis of the Effect of LoRA Rank in Section 5.2.3 may help. In this section, we compare the results of each method with different LoRA ranks, and it shows that a higher rank does not necessarily achieve better results. This means more trainable parameters do not result in higher performance, as a higher LoRA rank corresponds to more trainable parameters. We also present some results in Table A2. They show that the performance of FFA-LoRA with rank 16 is still inferior to that of FedSA-LoRA with rank 8 (ensuring they share the same number of trainable parameters).
> Moreover, our FedSA-LoRA shares the same trainable parameters with LoRA and has fewer trainable parameters compared to FedDPA-LoRA. However, the performance of FedSA-LoRA is still superior to them. This demonstrates that the superiority of FedSA-LoRA is not derived from more trainable parameters. We attribute the superiority of FedSA-LoRA to the personalization design of matrices $B$ and the global sharing of matrices $A$, which is motivated by Lemma 1 and supported by empirical validations.
>
> We hope this explanation can address the reviewer’s concerns.
>
> | Method | Trainable Parameters | Communication Parameters | QNLI ↑ | SST2 ↑ | MNLI-m ↑ |
> |-|-|-|-|-|-|
> | LoRA (r=8)           | 1.83M | 0.78M | 90.69  | 95.26  | 88.80    |
> | FFA-LoRA (r=8)      | 1.44M | 0.39M | 91.72  | 94.91  | 88.83    |
> | FFA-LoRA (r=16)     | 1.83M | 0.78M | 91.62  | 94.13  | 89.25    |
> | FedDPA-LoRA (r=8)   | 2.62M | 0.78M | 90.74  | 95.50  | 88.99    |
> | FedSA-LoRA (r=8)    | 1.83M | 0.39M | 92.00  | 95.92  | 90.20    |
>
> #### Table A2. Trainable and communication parameters, along with the corresponding performance for each method based on LoRA using the RoBERTa-large model.
>
> > **W7**: add more advanced baselines, such as FedLoRA [2], FedDPA [3], and HetLoRA [4].
>
> Thank you for the constructive comments. We have further added an advanced baseline, i.e., FedDPA [3], in our revision. As shown in Table A1 in our **Response to All** for the added comparison, our FedSA-LoRA outperforms FedDPA, further demonstrating the superiority of our approach.
>
> Please note that HetLoRA [4] is designed to handle that clients have heterogeneous ranks, which is not comparable to the setting in this paper. Additionally, FedDPA [3] has shown its superiority to FedLoRA [2] in their work. Thus, we did not provide a comparison with these two methods.  We hope the added advanced baseline can address the reviewers' concerns.
>
> [1] Jianyu Wang, Qinghua Liu, Hao Liang, Gauri Joshi, and H Vincent Poor. Tackling the objective inconsistency problem in heterogeneous federated optimization. Advances in neural information processing systems, 33:7611–7623, 2020.
>
> [2] Liping Yi, Han Yu, Gang Wang, and Xiaoguang Liu. Fedlora: Model-heterogeneous personalized federated learning with lora tuning. arXiv preprint arXiv:2310.13283, 2023.
>
> [3] Yiyuan Yang, Guodong Long, Tao Shen, Jing Jiang, and Michael Blumenstein. Dual-personalizing adapter for federated foundation models. arXiv preprint arXiv:2403.19211, 2024.
>
> [4] Yae Jee Cho, Luyang Liu, Zheng Xu, Aldi Fahrezi, Matt Barnes, and Gauri Joshi. Heterogeneous lora for federated fine-tuning of on-device foundation models. In International Workshop on Federated Learning in the Age of Foundation Models in Conjunction with NeurIPS 2023, 2023.
>
> [5] Li, Xiang, Kaixuan Huang, Wenhao Yang, Shusen Wang, and Zhihua Zhang. "On the convergence of fedavg on non-iid data." arXiv preprint arXiv:1907.02189, 2019.
>
> [6] Yu, Hao, Sen Yang, and Shenghuo Zhu. "Parallel restarted SGD with faster convergence and less communication: Demystifying why model averaging works for deep learning." Proceedings of the AAAI conference on artificial intelligence. Vol. 33. No. 01. 2019.
>
> [7] Basu, Debraj, et al. "Qsparse-local-SGD: Distributed SGD with quantization, sparsification and local computations." Advances in Neural Information Processing Systems 32, 2019.
>
> [8] Koloskova, Anastasia, Sebastian Stich, and Martin Jaggi. "Decentralized stochastic optimization and gossip algorithms with compressed communication." International Conference on Machine Learning. PMLR, 2019.

---

> ### Comment · Reviewer_KuVq · 2024-11-22
>
> I highly appreciate the authors' thorough explanations regarding Figure 2 and the computation/communication costs. The added baseline comparisons make sense to me as well. But I maintain my concerns about the novelty of the theoretical analysis, so I increase my rating to 5.

---

> ### Author Response · Authors · 2024-11-23
> **Thanks for Reply**
>
> Many thanks for the reviewer's reply. We are pleased that our response addressed most of the reviewer's concerns. We would like to provide further explanation regarding the reviewer's concern about the novelty of the theoretical analysis.
>
> To our understanding, the reviewer's concerns about the theoretical analysis may be due to Assumption 2 (bounded second moment assumption) used in our paper. Please correct us if we have misunderstood this point. Here, we would like to provide further explanation about this assumption. This assumption often holds when we have bounded variance in the samples of the training dataset, indicating that there is not much noise in our training data. This assumption is also widely used in many federated learning papers [1, 2, 3, 4] to derive their convergence rates. Under this same assumption, we can derive a similar convergence rate to traditional FedAvg. Additionally, **we would like to clarify that we do not claim the convergence rate analysis as one of our contributions**, though it is lacking in previous LoRA-based FL works [5, 6]; it exists in many traditional FL papers [1, 2, 3, 4].
>
> Moreover, we would like to emphasize that the major contribution of this work is the symmetry analysis of the $A$ and $B$ matrices within LoRA in the context of FL and the extension of this analysis to other LoRA variants. Based on our findings, we then introduce our method. Extensive evaluation and system efficiency analysis demonstrate the superiority of our method in terms of accuracy and efficiency.
>
> We hope this explanation addresses the reviewer's concern about the novelty of the theoretical analysis. Please feel free to reach out if the reviewer still has concerns, and we would be glad to discuss them with the reviewer.
>
> [1] Li, Xiang, et al. "On the convergence of fedavg on non-iid data." arXiv preprint arXiv:1907.02189, 2019.
>
> [2] Yu, Hao, et al. "Parallel restarted SGD with faster convergence and less communication: Demystifying why model averaging works for deep learning." AAAI, 2019.
>
> [3] Basu, Debraj, et al. "Qsparse-local-SGD: Distributed SGD with quantization, sparsification and local computations." NeurIPS, 2019.
>
> [4] Koloskova, Anastasia, et al. "Decentralized stochastic optimization and gossip algorithms with compressed communication." ICML, 2019.
>
> [5] Sun, Youbang, et al. "Improving loRA in privacy-preserving federated learning." ICLR, 2024.
>
> [6] Yang, Yiyuan, et al. "Dual-Personalizing Adapter for Federated Foundation Models." arXiv preprint arXiv:2403.19211 (2024).

---

> > ### Comment · Reviewer_KuVq · 2024-11-23
> >
> > Thanks for your further response, I would like to keep my current rating.

---

> > > ### Author Response · Authors · 2024-11-23
> > > **Thanks for Reply**
> > >
> > > Many thanks for the reviewer's reply. Please feel free to reach out if the reviewer has any further concerns, and we would be glad to discuss them with the reviewer.

---

### Author Response · Authors · 2024-11-22
**Response to All**

We thank all the reviewers for their constructive comments. Several important suggestions were made, and we have carefully considered each and revised the manuscript accordingly. Please find below a summary of the key updates that have been incorporated into the manuscript. All changes have been marked in blue in the revised manuscript.

1. **Comprehensively evaluate the effectiveness of the proposed method**:
- add an advanced baseline, FedDPA [1];
 - conduct multiple runs to report the mean and standard deviation;
 - include a system efficiency analysis;
 - add one more code generation task.

2. **Better present our work**:
 - more effectively motivate LoRA-based methods;
 - introduce personalization FL;
 - provide additional explanations about the assumptions used in our work and correct typos in Assumption 1;
 - add a discussion about related works on the asymmetry of $A$ and $B$ in LoRA;
 - include Figure 3 to separately show the relations of $A$ from Figure 2;
 - provide more explanation about aggregation errors;
 - explain why our method is better in non-IID settings.

Moreover, we further present our main results below, including multiple runs recommended by **Reviewers Lpwz and dqpx**, as well as the advanced baseline FedDPA [1] suggested by **Reviewer KuVq**.
The experimental results demonstrate the stability and effectiveness of the proposed method. We hope the added advanced baseline and multiple runs can address the reviewers' concerns.

|        | Method             | MNLI-m         | MNLI-mm        | SST2          | QNLI          | QQP           | RTE           | Avg.    |
|--------|--------------------|----------------|----------------|---------------|----------------|----------------|----------------|---------|
| **LoRA**  | LoRA               | $88.71_{\pm 0.09}$ | $88.19_{\pm 0.02}$ | $95.16_{\pm 0.09}$ | $91.16_{\pm 0.72}$ | $85.33_{\pm 1.33}$ | $87.49_{\pm 0.15}$ | $89.33$ |
|        | FFA-LoRA           | $88.83_{\pm 0.02}$ | $88.27_{\pm 0.03}$ | $94.95_{\pm 0.04}$ | $91.52_{\pm 0.59}$ | $86.71_{\pm 0.07}$ | $86.08_{\pm 1.16}$ | $89.39$ |
|        | FedDPA-LoRA        | $88.99_{\pm 0.06}$ | $88.43_{\pm 0.05}$ | $95.50_{\pm 0.06}$ | $90.74_{\pm 1.38}$ | $85.73_{\pm 1.73}$ | $87.44_{\pm 0.13}$ | $89.47$ |
|        | FedSA-LoRA         | **$90.18_{\pm 0.02}$** | **$88.88_{\pm 0.02}$** | **$96.00_{\pm 0.04}$** | **$92.13_{\pm 0.24}$** | **$87.48_{\pm 0.22}$** | **$87.93_{\pm 0.11}$** | **$90.43$** |
| **rsLoRA** | rsLoRA            | $88.91_{\pm 0.15}$ | $88.33_{\pm 0.04}$ | $95.02_{\pm 0.24}$ | $91.21_{\pm 0.39}$ | $86.73_{\pm 0.98}$ | $85.99_{\pm 0.34}$ | $89.36$ |
|        | FFA-rsLoRA        | $89.21_{\pm 0.11}$ | $88.45_{\pm 0.08}$ | $95.42_{\pm 0.17}$ | $91.42_{\pm 0.44}$ | $86.93_{\pm 1.18}$ | $85.24_{\pm 0.21}$ | $89.44$ |
|        | FedDPA-rsLoRA     | $89.34_{\pm 0.11}$ | $88.53_{\pm 0.05}$ | $95.56_{\pm 0.21}$ | $90.97_{\pm 0.76}$ | $86.81_{\pm 0.53}$ | $86.26_{\pm 0.11}$ | $89.57$ |
|        | FedSA-rsLoRA      | **$90.35_{\pm 0.11}$** | **$89.02_{\pm 0.03}$** | **$95.78_{\pm 0.08}$** | **$92.03_{\pm 0.22}$** | **$87.97_{\pm 0.16}$** | **$88.00_{\pm 0.10}$** | **$90.52$** |
| **VeRA**  | VeRA               | $85.54_{\pm 0.10}$ | $85.09_{\pm 0.07}$ | $93.53_{\pm 0.13}$ | $91.90_{\pm 0.17}$ | $82.07_{\pm 0.35}$ | $86.31_{\pm 0.12}$ | $87.40$ |
|        | FFA-VeRA          | $86.63_{\pm 0.13}$ | $86.22_{\pm 0.08}$ | $93.44_{\pm 0.05}$ | $92.05_{\pm 0.23}$ | $82.23_{\pm 0.07}$ | $83.54_{\pm 0.59}$ | $87.35$ |
|        | FedDPA-VeRA       | $86.74_{\pm 0.11}$ | $86.35_{\pm 0.07}$ | $93.61_{\pm 0.32}$ | $90.73_{\pm 0.54}$ | $82.11_{\pm 0.41}$ | $86.12_{\pm 0.12}$ | $87.61$ |
|        | FedSA-VeRA        | **$87.21_{\pm 0.10}$** | **$86.52_{\pm 0.04}$** | **$93.68_{\pm 0.07}$** | **$92.91_{\pm 0.09}$** | **$82.56_{\pm 0.05}$** | **$87.83_{\pm 0.09}$** | **$88.45$** |
#### Table A1. Performance of different methods on the GLUE benchmark. MNLI-m denotes MNLI with matched test sets, and MNLI-mm denotes MNLI with mismatched test sets. For all tasks, we report accuracy evaluated across 3 runs with mean and standard deviation.

[1] Yang, Yiyuan, et al. "Dual-Personalizing Adapter for Federated Foundation Models." arXiv preprint arXiv:2403.19211 (2024).

---

### Meta-Review · Area_Chair_NCeQ · 2024-12-18

**Metareview:**

**Summary:** The authors show that in an FL setting, the A and B matrices have different roles for LoRA. Specifically, the B matrices learn more client-specific features while the A matrices are responsible for general knowledge. Based on this observation, the A matrices are shared at the server for aggregation and the B matrices are client-specific. This reduces the communication cost and potentially leads to learning better features.

**Strengths:** The observation that the A and B matrices have different roles in LoRA FL is interesting. The authors perform thorough analysis and ablations. The proposed method tackles an important problem and shows improvements over the previous approaches. The authors provide some theoretical analysis as well.

**Weaknesses:** Some reviewers raise concerns about the novelty of the A and B matrices, specifically, the asymmetry has been observed before in earlier work. The reviewers also claim some of the assumptions for proving the bounds might be too strong. Some of the results lacked confidence intervals, which the authors provided during the rebuttal.

**Decision:** Initially, there were several concerns about the assumptions, novelty, and validity of the experimental results. The authors did a decent job of addressing these concerns by providing thorough feedback and additional experimental results during the discussion period. As a result of such improvements, the reviewers raised their scores and the overall response toward the paper is positive. Thus, I recommend acceptance.

**Additional Comments On Reviewer Discussion:**

The authors provided additional experimental evidence during the rebuttal period. Additionally, they clarified some of the concerns about the theoretical assumptions and the aggregation errors. Overall, their response was thorough and satisfactory to the majority of the reviewers.

---

### Decision · Program_Chairs · 2025-01-22

Accept (Poster)